# Rethinking KL Regularization in RLHF:
# From Wrong Value Estimation to Correct Gradient Optimization

## Abstract

Reinforcement Learning from Human Feedback (RLHF) leverages a Kullback-Leibler (KL) divergence loss to stabilize training and prevent overfitting. However, in methods such as GRPO, its implementation may be guided by principles from numerical value estimation — a practice that overlooks the term's functional role as an optimization loss. To analyze this issue, we establish a unified framework that connects two seemingly distinct implementation styles: using the mathematical term $k_n$ as a detached coefficient for the policy's score function ($k_n$ **in reward**) or as a direct loss function through which gradients are propagated ($k_n$ **as loss**). We show that the latter can always be analyzed via an equivalent gradient coefficient in the former, unifying the two perspectives. Through this framework, we first prove that conclusions from value estimation fail to guide proper KL loss design, using the $k_1$ **as loss** as a counterexample. We then prove the conventional $k_1$ **in reward** (like PPO) is the principled loss for Reverse KL (RKL) regularization. We further establish a key finding: under on-policy conditions, the $k_2$ **as loss** formulation is, in fact, gradient-equivalent to $k_1$ **in reward**. This equivalence, first proven in our work, identifies both as the theoretically sound implementations of the RKL objective. In contrast, we show that the recently adopted $k_3$ **as loss** (like GRPO) is merely a first-order, biased approximation of the principled loss. Furthermore, we argue that common off-policy implementations of $k_n$ **as loss** methods are biased due to neglected importance sampling, and we propose a principled correction. Our findings provide a comprehensive, gradient-based rationale for choosing and correctly implementing KL regularization, paving the way for more robust and effective RLHF systems.

---

[1]. Correspondence to: Anonymous Author.

## 1. Introduction

Training state-of-the-art Large Language Models (LLMs) is a multi-stage process. Following large-scale pretraining and Supervised Fine-Tuning (SFT), a final post-training stage, Reinforcement Learning from Human Feedback (RLHF), is often employed to align models with preferences and to improve performance on demanding tasks such as mathematical reasoning (Ouyang et al., 2022; Shao et al., 2024). A core component of RLHF is **KL regularization**: a penalty derived from the Kullback–Leibler divergence (Kullback & Leibler, 1951) that constrains the policy to stay close to a reference model (typically the SFT checkpoint), thereby improving training stability and reducing reward overfitting (Ouyang et al., 2022; Stiennon et al., 2020).

Despite the critical role of the KL loss, its theoretical foundations in optimization remain underexplored. Across algorithms and codebases, the KL penalty takes several subtly different forms, and a common justification is how well a single-sample term estimates the KL *value* (e.g., bias/variance). However, in RLHF, the KL term serves as an *optimization loss*, not merely a diagnostic: because the sampling distribution depends on the policy parameters, good value estimation does not guarantee that the induced gradient matches the intended regularizer. As a result, implementations that look nearly identical in code—often differing only in whether a term is detached or backpropagated—can correspond to qualitatively different update directions and stability behavior.

This paper argues that a gradient-centric perspective is essential for designing robust and effective RLHF algorithms. We establish a unified framework that connects two common implementation styles: using the term $k_n$ as a detached coefficient ($k_n$ **in reward**) versus differentiating a surrogate penalty directly ($k_n$ **as loss**). This framework allows us to compare implementations through the scalar coefficient they induce on the score function $\nabla_{\boldsymbol{\theta}} \log \pi_{\boldsymbol{\theta}}$.

Our main contributions are threefold:

1. **A gradient-correctness criterion for KL loss design.** We show that conclusions from KL value estimation do not necessarily translate into correct optimization be-

havior. In particular, $k_1$ **as loss** provides a clean counterexample: despite $k_1$ being an unbiased KL value estimator, it has zero expected gradient under on-policy sampling and thus provides no KL constraint.

2. **Identifying principled RKL implementations.** We derive the exact on-policy Reverse KL (RKL) gradient and prove that $k_1$ **in reward** (PPO-style) and $k_2$ **as loss** are gradient-equivalent, establishing both as theoretically sound choices for RLHF regularization.

3. **Diagnosing GRPO-style surrogates and off-policy pitfalls.** We analyze $k_3$ **as loss** (used in GRPO) as a biased first-order surrogate with mismatched tail behavior. We further identify missing importance-sampling and clipping corrections as a key source of bias when "as loss" heads are reused off-policy, and provide a principled correction.

## 2. Related Work

Our work intersects three lines of research: KL value estimation, RLHF systems, and KL loss design. We briefly review each and clarify how our gradient-centric perspective complements prior contributions.

**KL Value Estimation.** The KL divergence is often estimated via Monte Carlo sampling, motivating a family of single-sample terms $k_1, k_2, k_3$ (John, 2020). Prior discussions evaluate them as *value estimators*, characterizing $k_1$ as unbiased but high-variance, $k_2$ as biased but lower-variance, and $k_3$ as an "optimal" low-variance unbiased estimator under regularity assumptions. We complement this view in two ways: we show that the advertised superiority of $k_3$ can fail under distribution shift and heavy-tailed importance ratios (Appendix I); and more importantly, we argue that value-estimation criteria are insufficient when these terms are used as *optimization losses*, where the induced gradient is what determines the regularization effect.

**RLHF Systems and Methods.** Open-source RLHF frameworks (e.g., OpenRLHF (Hu et al., 2024), Verl (Sheng et al., 2024), slime (Zhu et al., 2025), and ROLL (Wang et al., 2025)) support PPO-style updates (Ouyang et al., 2022) and more recent critic-free variants. They emphasize systems-level challenges such as efficient rollouts (e.g., OpenRLHF uses vLLM (Kwon et al., 2023)) and actor–critic stability. Several recent works focus on the critic bottleneck: VAPO (Yue et al., 2025) proposes critic pretraining, while GRPO and Reinforce++ remove the critic altogether to enable larger actor scaling. Most methods retain KL regularization, though some rule-based reward algorithms (e.g., DAPO) propose removing it for performance; ProRL (Liu et al., 2025b) mitigates this trade-off by periodically resetting the reference model, highlighting that KL remains important for long-horizon stability. Our contribution is complementary: we analyze the gradient correctness of the KL regularizer itself, which impacts stability regardless of the surrounding loss.

**KL Loss in RLHF.** The practice of adding a KL penalty *in the reward* traces back to InstructGPT (Ouyang et al., 2022), where the log-ratio acts as a detached coefficient multiplying the policy score function. Jaques et al. (2019) noted a potential connection between adding KL to the reward versus to the loss, but without a formal gradient-level analysis. More recently, GRPO (Shao et al., 2024) (adopted in DeepSeek-R1 (Guo et al., 2025)) popularized directly differentiating through a $k_3$-style surrogate, often justified by its value-estimation properties (John, 2020). Our work provides a unified framework to compare these choices at the gradient level, identifies when they match the intended Reverse KL objective, and discusses connections to Forward KL in Appendix K.

## 3. Preliminaries

Before analyzing KL implementations, we introduce the necessary background: the single-sample terms commonly used to approximate KL divergences, and the standard RLHF objective they are meant to regularize.

### 3.1. Single-Sample Terms for KL Estimation

The Kullback–Leibler (KL) divergence from a distribution $q(x)$ to a reference $p(x)$ is defined as

$$D_{\text{KL}}(q \parallel p) = \mathbb{E}_{x \sim q} \left[ \log \frac{q(x)}{p(x)} \right]. \qquad (1)$$

When this expectation is intractable, practitioners estimate it from Monte Carlo samples. Defining the likelihood ratio $\delta(x) = p(x)/q(x)$, common single-sample terms include (John, 2020):

$$k_1(x) = -\log \delta(x),$$
$$k_2(x) = \tfrac{1}{2} \left( \log \delta(x) \right)^2,$$
$$k_3(x) = \delta(x) - 1 - \log \delta(x).$$

In RLHF, we instantiate these terms with $q(\cdot) = \pi_\theta(\cdot|x)$ (the current policy) and $p(\cdot) = \pi_{\text{ref}}(\cdot|x)$ (the reference). The term $k_3$ is designed to be nonnegative and to reduce variance when $p$ and $q$ are close; however, the "strictly better estimator" claim (John, 2020) relies on regularity conditions that can fail under distribution shift (see Appendix I for counterexamples). More fundamentally, when these terms are used for *regularization*, their suitability depends on the induced gradient, not on value-estimation properties—a distinction we formalize in the next section. In particular, when the support or tails of $p$ and $q$ differ significantly,

$k_3$ can exhibit heavy-tailed behavior: its sample mean can be dominated by rare events and its variance may become extremely large (or even infinite) without additional assumptions. Therefore, using $k_3$ as a "low-variance unbiased" value estimator requires verifying conditions that are often violated in practice; Appendix I provides counterexamples and discussion.

> **Gap in prior work.** The blog (John, 2020) cited by GRPO to justify $k_3$ discusses only *value estimation*—not *loss design*. Even as a value estimator, $k_3$ can fail under distribution shift (Appendix I).

### 3.2. The RLHF Objective

RLHF fine-tunes a policy $\pi_{\boldsymbol{\theta}}$ to produce responses $y$ to prompts $x$ that maximize a learned reward $r(x, y)$. Pure reward maximization can lead to reward hacking and drift from a trusted SFT policy. A standard remedy is to regularize toward a fixed reference $\pi_{\text{ref}}$ via the Reverse KL (RKL):

$$\begin{aligned} \mathcal{J}_{\text{RLHF}}(\boldsymbol{\theta}) &= \mathbb{E}_{x \sim \mathcal{D}}\, \mathbb{E}_{y \sim \pi_{\boldsymbol{\theta}}(\cdot|x)}\big[r(x, y)\big] \\ &\quad - \beta\, D_{\text{KL}}(\pi_{\boldsymbol{\theta}}(\cdot|x) \,\|\, \pi_{\text{ref}}(\cdot|x)) \\ &\triangleq \mathcal{J}_{\text{Reward}}(\boldsymbol{\theta}) - \beta\, \mathcal{J}_{\text{RKL}}(\boldsymbol{\theta}). \end{aligned} \quad (2)$$

Here $\mathcal{D}$ is the prompt distribution, $\beta > 0$ controls the strength of regularization, and in practice $y$ is sampled from a detached snapshot $\pi_\theta$ that is numerically equal to $\pi_{\boldsymbol{\theta}}$ at sampling time.

The high-level objective is clear, but it leaves open an important implementation choice: *how should the KL term be integrated into the policy-gradient update?* Different choices lead to different induced gradients. We address this question next.

# 4. A Unified Framework for KL Regularization

Although most RLHF algorithms optimize the same high-level objective (Equation (2)), their concrete implementations of the KL term differ. This section introduces a unified framework that makes these differences explicit and comparable.

The key observation is that any practical KL implementation, regardless of its form, ultimately induces a scalar coefficient $c(x, y)$ multiplying the policy score function $\nabla_{\boldsymbol{\theta}} \log \pi_{\boldsymbol{\theta}}(y|x)$. This coefficient determines both the expected update direction and the variance of the stochastic gradient. By making $c(x, y)$ explicit, we can directly compare implementations: a KL loss is *gradient-correct* if and only if its induced coefficient matches that of the target KL objective.

**Notation.**

> **Key convention (red vs. black).** We use $\boldsymbol{\theta}$ (red) to denote trainable parameters that carry gradients, while $\theta$ (black) denotes a detached snapshot (no gradients).

Samples $y$ are drawn from the detached snapshot $\pi_\theta(\cdot|x)$, which is numerically equal to $\pi_{\boldsymbol{\theta}}(\cdot|x)$ at sampling time. Scalar coefficients multiplying the score function are always treated as detached.[1]

### 4.1. Core Components of the RLHF Objective

The practical RLHF objective consists of two terms—reward maximization and KL regularization—both estimated via Monte Carlo sampling.

**Reward Maximization.** The primary goal is to maximize expected reward: $\mathcal{J}_{\text{Reward}}(\boldsymbol{\theta}) = \mathbb{E}_{x \sim \mathcal{D},\, y \sim \pi_{\boldsymbol{\theta}}(\cdot|x)}[r(x, y)]$. Since sampling from $\pi_{\boldsymbol{\theta}}$ is non-differentiable, the gradient is estimated via the log-derivative trick (REINFORCE (Williams, 1992); see Appendix B):

$$\nabla_{\boldsymbol{\theta}} \mathcal{J}_{\text{Reward}}(\boldsymbol{\theta}) = \mathbb{E}_{x \sim \mathcal{D},\, y \sim \pi_{\boldsymbol{\theta}}(\cdot|x)}\left[r(x, y) \cdot \nabla_{\boldsymbol{\theta}} \log \pi_{\boldsymbol{\theta}}(y|x)\right]. \quad (3)$$

Here $r(x, y)$ is typically a shaped advantage signal rather than a raw reward (details in Appendix A).

**KL Regularization.** The RKL term constrains the policy toward the reference:

$$\begin{aligned} \mathcal{J}_{\text{RKL}}(\boldsymbol{\theta}) &= \mathbb{E}_{x \sim \mathcal{D}}\Big[D_{\text{KL}}\big(\pi_{\boldsymbol{\theta}}(\cdot|x) \,\|\, \pi_{\text{ref}}(\cdot|x)\big)\Big] \\ &= \mathbb{E}_{x \sim \mathcal{D},\, y \sim \pi_{\boldsymbol{\theta}}(\cdot|x)}\Big[\log \pi_{\boldsymbol{\theta}}(y|x) - \log \pi_{\text{ref}}(y|x)\Big]. \end{aligned} \quad (4)$$

Like the reward term, practical implementations estimate this expectation with samples $y \sim \pi_\theta(\cdot|x)$ and differentiate only through $\pi_{\boldsymbol{\theta}}$ (see Appendix C). However, *how* the KL-derived term interacts with the gradient varies across algorithms. We distinguish two common styles:

1. $k_n$ **in reward (detached coefficient):** The KL-derived scalar $k_n$ multiplies the score function as a *detached* coefficient—gradients do not flow through $k_n$ itself. This is the style used in PPO, with $k_1 = \log \pi_\theta - \log \pi_{\text{ref}}$:

$$\begin{aligned} \mathcal{J}_{k_n \text{ in reward}}(\boldsymbol{\theta}) := \mathbb{E}_{x \sim \mathcal{D},\, y \sim \pi_{\boldsymbol{\theta}}(\cdot|x)} \\ \Big[k_n\big(\pi_\theta(y \mid x),\, \pi_{\text{ref}}(y \mid x)\big) \log \pi_{\boldsymbol{\theta}}(y \mid x)\Big]. \end{aligned} \quad (5)$$

2. $k_n$ **as loss (direct differentiation):** The KL-derived term is treated as a standalone loss, with gradients

---

[1] We treat the entire response $y$ as a single action, working with the joint probability $\pi(y|x)$ rather than token-level factors. This bandit-style simplification suffices for analyzing the core gradient properties.

propagated *through* it:

$$k_3 = \frac{\pi_{\text{ref}}}{\pi_{\boldsymbol{\theta}}} - \log \frac{\pi_{\text{ref}}}{\pi_{\boldsymbol{\theta}}} - 1 \quad \text{(GRPO)}, \quad k_2 = \frac{1}{2}\left(\log \frac{\pi_{\boldsymbol{\theta}}}{\pi_{\text{ref}}}\right)^2 \text{(Ours)}.$$

The corresponding objective is:

$$\mathcal{J}_{k_n \text{ as loss}}(\boldsymbol{\theta}) := \mathbb{E}_{x\sim\mathcal{D},\, y\sim\pi_{\theta}(\cdot|x)}\left[k_n\big(\pi_{\boldsymbol{\theta}}(y|x),\, \pi_{\text{ref}}(y|x)\big)\right]. \quad (6)$$

Both forms are *optimization surrogates*: their scalar values need not equal a KL divergence; what matters is the gradient they induce.

**Connecting the two styles.** Although $k_n$ as loss differentiates through $k_n$ directly, its gradient can always be written in the $k_{n'}$ in reward form for some equivalent coefficient $k_{n'}$. This coefficient is obtained by differentiating $k_n$ with respect to the log-policy:

**Equivalent coefficient formula:**

$$k_{n'} := \frac{\partial k_n}{\partial \log \pi_{\boldsymbol{\theta}}}\bigg|_{\pi_{\boldsymbol{\theta}}=\pi_{\theta}}. \quad (7)$$

This formula allows direct comparison of the two styles. A key example: $k_{2'} = k_1$. In Section 5.2, we prove that $k_1$ in reward and $k_2$ as loss are therefore gradient-equivalent.

### 4.2. Integration Forms and Algorithm Mapping

The KL formulation also determines how it is integrated with the reward objective in practice.

**Combined vs. Decoupled Forms.** Since $k_n$ in reward shares the same score function as the reward term, its coefficient can be merged to form a single "advantage":

$$\mathcal{L}_{\text{Combined}}(\boldsymbol{\theta}) = -\mathbb{E}_{x,y}\Big[A_{\text{comb}}(x,y)\,\log \pi_{\boldsymbol{\theta}}(y|x)\Big], \quad (8)$$
$$A_{\text{comb}}(x,y) := r(x,y) - \beta\,k_n.$$

In contrast, $k_n$ as loss naturally yields a **decoupled** objective with a separate KL head:

$$\mathcal{L}_{\text{Decoupled}}(\boldsymbol{\theta}) = -\mathbb{E}_{x\sim\mathcal{D},\, y\sim\pi_{\theta}}\Big[r(x,y)\cdot\log\pi_{\boldsymbol{\theta}}(y|x)\Big]$$
$$+ \beta\,\mathbb{E}_{x\sim\mathcal{D},\, y\sim\pi_{\theta}}\Big[k_n\big(\pi_{\boldsymbol{\theta}}(y|x),\pi_{\text{ref}}(y|x)\big)\Big]. \quad (9)$$

**Off-policy considerations.** In off-policy updates (e.g., PPO with multiple gradient steps per rollout), the combined form inherits importance-sampling (IS) and clipping corrections via the standard PPO surrogate $\pi_{\boldsymbol{\theta}}/\pi_{\theta_k}$. In contrast, $k_n$ as loss implementations must apply IS and clipping to the KL head itself—a step commonly omitted in practice, leading to biased gradients (see Appendix G).

**Positioning PPO and GRPO.** Table 1 summarizes where common algorithms fall in this taxonomy. PPO uses $k_1$ **in reward** in a combined form; GRPO uses $k_3$ **as loss** in a decoupled form.

With the framework in place, we now analyze which KL losses are gradient-correct for the RKL objective and diagnose failure modes when they are not.

## 5. Gradient-Based Analysis of KL Implementations

We now apply the coefficient framework to analyze common KL implementations. The analysis proceeds in three steps: (i) a counterexample showing that $k_1$ **as loss** provides no regularization signal; (ii) derivation of the principled RKL gradient and its equivalent surrogates; and (iii) diagnosis of $k_3$ **as loss** as a biased first-order approximation.

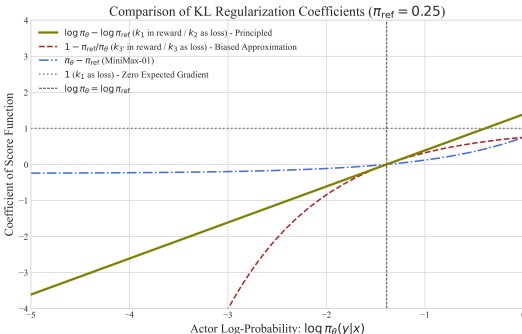

*Figure 1.* Induced score-function coefficients as a function of $\log \pi_{\theta}(y|x)$ (with $\pi_{\text{ref}} = 0.25$). The principled RKL implementations ($k_1$ **in reward** and $k_2$ **as loss**) yield coefficient $k_1$. The GRPO-style $k_3$ **as loss** uses coefficient $1 - \delta$ (where $\delta = \pi_{\text{ref}}/\pi_{\theta}$), a first-order approximation with mismatched tails. The direct $k_1$ **as loss** yields a constant 1, producing zero-mean noise.

### 5.1. Counterexample: Why $k_1$ as loss Fails

We begin with a counterexample that illustrates the gap between value estimation and optimization. Although $k_1 = \log(\pi_{\theta}/\pi_{\text{ref}})$ is an unbiased estimator of the KL value, using it directly as a loss provides no regularization.

Consider the direct-loss formulation with on-policy sampling:

$$\mathcal{J}_{k_1 \text{ as loss}}(\boldsymbol{\theta}) = \mathbb{E}_{x\sim\mathcal{D},\, y\sim\pi_{\theta}(\cdot|x)}\big[\log \pi_{\boldsymbol{\theta}}(y|x) - \log \pi_{\text{ref}}(y|x)\big]. \quad (10)$$

Since $\pi_{\text{ref}}$ does not depend on $\boldsymbol{\theta}$, differentiating yields

$$\nabla_{\boldsymbol{\theta}}\mathcal{J}_{k_1 \text{ as loss}}(\boldsymbol{\theta}) = \mathbb{E}_{x\sim\mathcal{D},\, y\sim\pi_{\theta}(\cdot|x)}\big[\nabla_{\boldsymbol{\theta}}\log \pi_{\boldsymbol{\theta}}(y|x)\big]. \quad (11)$$

The gradient is independent of $\pi_{\text{ref}}$—it carries *no* directional signal toward the reference. Moreover, by the zero-mean score identity (Lemma C.2), this expectation equals zero. In finite-sample (minibatch) practice, the gradient is

*Table 1.* Taxonomy of RLHF KL implementations. $k_n$ in reward can also be used in a decoupled form.

| Algorithm | Typical $k_n$ | Style | Form | Off-Policy Notes |
|---|---|---|---|---|
| PPO / REINFORCE | $k_1$ | in reward | Combined | IS/clipping inherited from PPO surrogate. |
| GRPO | $k_3$ | as loss | Decoupled | Requires explicit IS/clipping; often omitted. |

nonzero but has zero mean; **$k_1$ as loss** thus injects reference-independent noise that increases variance and can destabilize training, without providing any meaningful constraint.

This counterexample underscores a key point: good value estimation does not imply good optimization. A KL regularizer must be evaluated by its induced gradient, not its scalar estimate.

### 5.2. Principled RKL Surrogates: $k_1$ in reward $\Leftrightarrow k_2$ as loss

Having seen a failure mode, we now derive the gradient that a principled KL regularizer *should* produce. Applying the product rule and log-derivative trick to the RKL objective (Equation (4); see Appendix C) gives:

$$\nabla_{\boldsymbol{\theta}} \mathcal{J}_{\mathrm{RKL}}(\boldsymbol{\theta}) = \mathbb{E}_{x \sim \mathcal{D}}\left[ \sum_y \nabla_{\boldsymbol{\theta}} \pi_\theta(y|x) \left( \log \frac{\pi_\theta(y|x)}{\pi_{\mathrm{ref}}(y|x)} + 1 \right) \right]. \quad (12)$$

By the zero-mean score identity in Lemma C.2, the term '+1' vanishes in expectation, resulting in the practical form of the policy gradient:

**Target RKL Gradient (the "gold standard"):**

$$\nabla_{\boldsymbol{\theta}} \mathcal{J}_{\mathrm{RKL}}(\boldsymbol{\theta}) = \mathbb{E}_{x \sim \mathcal{D}, y \sim \pi_\theta}\Big[ \underbrace{(\log \pi_\theta - \log \pi_{\mathrm{ref}})}_{\text{coefficient } k_1} \underbrace{\nabla_{\boldsymbol{\theta}} \log \pi_\theta}_{\text{score function}} \Big] \quad (13)$$

Any surrogate that correctly implements RKL must reproduce this target gradient. The following theorem shows that two seemingly different designs achieve this.

**Theorem 5.1** (Gradient equivalence of principled RKL surrogates)**.** *Under on-policy sampling $y \sim \pi_\theta(\cdot|x)$, the following two objectives are **gradient-equivalent**—both produce Equation (13):*

$$\mathcal{J}_{k_1 \text{ in reward}} = \mathbb{E}\Big[ \underbrace{(\log \pi_\theta - \log \pi_{ref})}_{\text{coeff. } k_1 \text{ (no grad)}} \cdot \log \pi_{\boldsymbol{\theta}} \Big] \quad (14)$$

$$\mathcal{J}_{k_2 \text{ as loss}} = \mathbb{E}\Big[ \tfrac{1}{2} \underbrace{(\log \pi_{\boldsymbol{\theta}} - \log \pi_{ref})^2}_{\text{gradients flow through}} \Big] \quad (15)$$

**Key insight.** In Equation (14), $k_1$ is a fixed scalar—gradients flow only through $\log \pi_{\boldsymbol{\theta}}$. In Equation (15), gradients flow through the squared term. Yet differentiating the

square produces exactly $k_1$ as the coefficient:

$$\nabla_{\boldsymbol{\theta}} \tfrac{1}{2}(\log \tfrac{\pi_\theta}{\pi_{\mathrm{ref}}})^2 = \underbrace{(\log \tfrac{\pi_\theta}{\pi_{\mathrm{ref}}})}_{\text{same } k_1!} \cdot \nabla_{\boldsymbol{\theta}} \log \pi_\theta. \quad (16)$$

Thus, both forms induce the same update: $k_1 \times$ score function. See Appendix C for the full proof.

**Why prefer $k_2$ as loss?** While gradient-equivalent to **$k_1$ in reward**, the squared form $k_2 = \frac{1}{2}(\log \pi_\theta/\pi_{\mathrm{ref}})^2$ offers practical advantages: (i) it is *always non-negative*, avoiding sign cancellation that increases variance in $k_1$; (ii) its gradient coefficient $k_1$ emerges naturally from differentiation, making the implementation straightforward; and (iii) unlike $k_3$, its variance does not depend on the chi-squared divergence and remains well-behaved under moderate distribution shift. These properties make **$k_2$ as loss** a robust default for on-policy RKL regularization.

Off-policy updates require additional IS and clipping corrections (Appendix G).

### 5.3. The GRPO Surrogate: $k_3$ as loss as a First-Order Approximation

We now turn to **$k_3$ as loss**, popularized by GRPO:

$$\mathcal{J}_{k_3 \text{ as loss}}(\boldsymbol{\theta}) = \mathbb{E}_{x \sim \mathcal{D}, y \sim \pi_\theta(\cdot|x)}\left[ \frac{\pi_{\mathrm{ref}}}{\pi_{\boldsymbol{\theta}}} - \log \frac{\pi_{\mathrm{ref}}}{\pi_{\boldsymbol{\theta}}} - 1 \right]. \quad (17)$$

Differentiating yields the induced coefficient $k_{3'}$:

$$\nabla_{\boldsymbol{\theta}} \mathcal{J}_{k_3 \text{ as loss}}(\boldsymbol{\theta}) = \mathbb{E}_{x \sim \mathcal{D}, y \sim \pi_\theta(\cdot|x)}$$
$$\Big[ \underbrace{\left( 1 - \frac{\pi_{\mathrm{ref}}(y|x)}{\pi_\theta(y|x)} \right)}_{k_{3'} \text{ coefficient}} \nabla_{\boldsymbol{\theta}} \log \pi_\theta(y \mid x) \Big]$$
$$= \nabla_{\boldsymbol{\theta}} \mathcal{J}_{k_{3'} \text{ in reward}}(\boldsymbol{\theta}). \quad (18)$$

Let $\delta = \pi_{\mathrm{ref}}/\pi_\theta$. The principled RKL coefficient and the $k_3$-induced coefficient are:

$$c^\star = -\log \delta, \qquad c_{3'} = 1 - \delta. \quad (19)$$

Around $\delta = 1$, Taylor expansion gives $-\log \delta \approx 1 - \delta + \mathcal{O}((\delta - 1)^2)$, so $c_{3'}$ is a first-order approximation of $c^\star$. The mismatch beyond first order leads to three issues (see Appendix E for formal statements and Figure 1 for visualization):

1. **Bias.** For $\delta \neq 1$, the update direction differs from the true RKL gradient.

2. **Mismatched tails.** When $\pi_\theta > \pi_{\mathrm{ref}}$ ($\delta \to 0$), the principled coefficient $-\log \delta \to +\infty$ provides a strong restoring force, whereas $1 - \delta \to 1$ saturates—weakening the constraint precisely when drift is large. Conversely, when $\pi_\theta < \pi_{\mathrm{ref}}$ ($\delta \to \infty$), $1 - \delta \to -\infty$ diverges faster than $-\log \delta$, inducing potentially unbounded updates.

3. **Variance sensitivity.** $\mathrm{Var}[1 - \delta] = \chi^2(\pi_{\mathrm{ref}} \| \pi_\theta)$, the chi-square divergence, which can be unstable under distribution shift.

Appendix K provides an alternative view: $c_{3'} = 1 - \delta$ equals the Forward KL coefficient $-\delta$ plus an implicit baseline, making $k_3$ **as loss** a variance-reduced FKL estimator rather than an RKL surrogate.

### 5.4. Summary of Recommendations

Table 2 summarizes the key practical consequences of our analysis. The "Coefficient" column shows the scalar $c$ multiplying the score function $\nabla_\theta \log \pi_\theta$; a method is RKL-correct if $c = k_1$. Based on our analysis, we offer the following practical guidance (see Appendix G and Appendix F for details on the last two points):

- **Avoid $k_1$ as loss.** Its expected gradient is zero and independent of the reference—it provides no regularization, only noise.

- **Use $k_1$ in reward or $k_2$ as loss for principled RKL regularization.** These are gradient-equivalent under on-policy sampling and correctly implement the RKL objective.

- **Recognize $k_3$ as loss as a biased surrogate.** It can impose weaker constraints when drift is large and may induce unbounded updates in the opposite tail.

- **Apply IS/clipping in off-policy settings.** When using $k_n$ as loss with PPO-style updates, explicit importance sampling and clipping are required for the KL head; omitting them introduces systematic bias.

- **Consider bounded alternatives if stability is paramount.** The MSE-based penalty (e.g., MiniMax-01 loss) induces a coefficient bounded in $[-1, 1]$; see Appendix F.

We empirically validate these predictions in Section 6.

## 6. Experimental Validation

We validate our gradient analysis with controlled RLHF experiments on a mathematical reasoning task. The experiments address two questions derived from Section 5: (i)

Does $k_1$ **as loss** provide any regularization beyond noise? (ii) Does the principled $k_2$ **as loss** enforce a stronger constraint than its first-order surrogate $k_3$ **as loss**? We report 7B-scale results here; 1.5B-scale dynamics and downstream benchmarks appear in Appendix M and N.

### 6.1. Setup

**Dataset.** We use a curated subset of OpenR1-Math-220k,[2] comprising NuminaMath 1.5 prompts with reference solutions verified by Math-Verify.[3] After removing sequences exceeding 2048 tokens, 7,300 prompts remain.

**Training Configuration.** To isolate gradient effects, we use fully on-policy training: rollout batch size 32, 8 responses per prompt, update batch size 256, and sampling temperature 1.0. The actor is Qwen2.5-Math-7B (Yang et al., 2024); rewards combine regex-based format checking and Math-Verify accuracy. We disable entropy regularization. Unless otherwise stated, we set the KL weight to $\beta = 0.5$; to better reflect practical settings with weaker constraints, we additionally report diagnostics at $\beta = 0.001$ (which permits larger drift and makes instabilities easier to observe). A numerical stability issue with group normalization and our mitigation are discussed in Appendix J.

### 6.2. Results

We track reward, accuracy, KL and log-probability gaps to the reference, group-level reward standard deviation, and response length.

**Experiment 1: $k_1$ as loss vs. no KL.** Figure 2 compares $k_1$ **as loss** to a no-KL baseline. As derived in Section 5, the gradient of $k_1$ **as loss** is independent of $\pi_{\mathrm{ref}}$ and has zero expectation; it should behave similarly to no KL, contributing only variance. Indeed, the two runs are comparable in reward/accuracy, but $k_1$ **as loss** exhibits larger KL and log-probability gaps at later stages, consistent with added stochasticity rather than regularization. Notably, both the no-KL and $k_1$ **as loss** settings reach higher reward than runs with effective KL constraints (Figure 3), reinforcing that $k_1$ **as loss** does not meaningfully limit drift.

**Experiment 2: $k_2$ as loss vs. $k_3$ as loss.** Figure 3 compares the principled $k_2$ **as loss** to the GRPO-style $k_3$ **as loss**. As a first-order approximation, $k_3$ **as loss** is asymmetric: it penalizes more strongly than $k_2$ **as loss** when $\pi_\theta < \pi_{\mathrm{ref}}$ but saturates when $\pi_\theta > \pi_{\mathrm{ref}}$. Since RL training typically drives the policy toward higher-reward regions

---

[2] https://huggingface.co/datasets/open-r1/OpenR1-Math-220k

[3] https://github.com/huggingface/Math-Verify

*Table 2.* KL regularization selection guide. Target coefficient for RKL: $c^\star = k_1 = \log(\pi_\theta/\pi_{\mathrm{ref}})$. Let $\delta = \pi_{\mathrm{ref}}/\pi_\theta$.

| Implementation | Coeff. | RKL? | Off-policy | Recommendation |
|---|---|---|---|---|
| **In-reward form**: $\mathcal{L} = \mathbb{E}[k_n \cdot \log \pi_{\boldsymbol{\theta}}]$ | | (coefficient $k_n$ does not carry gradients) | | |
| $\boldsymbol{k_1}$ **in reward** | $k_1$ | ✓ | Inherits PPO clip | **Recommended.** Principled RKL. |
| $k_2$ in reward | $k_2$ | ✗ | – | Avoid. Wrong coefficient. |
| $k_3$ in reward | $k_3$ | ✗ | – | Avoid. Mixed RKL−FKL. |
| MiniMax-01 | $\pi_\theta - \pi_{\mathrm{ref}}$ | – | Inherits PPO clip | Bounded $[-1, 1]$; MSE-based. |
| **As-loss form**: $\mathcal{L} = \mathbb{E}[k_n(\pi_{\boldsymbol{\theta}}, \pi_{\mathrm{ref}})]$ | | (gradients flow through $k_n$) | | |
| $k_1$ as loss | 1 | ✗ | – | **Avoid.** Zero expected gradient. |
| $\boldsymbol{k_2}$ **as loss** | $k_1$ | ✓ | Needs IS/clip$^\dagger$ | **Recommended.** Low-variance, stable. |
| $k_3$ as loss | $1-\delta$ | ≈ | Needs IS/clip$^\dagger$ | Biased 1st-order; mismatched tails. |
| DeepSeek-V3.2 | $\rho \cdot k_1$ | ✓* | Built-in IS | Equivalent to $k_1$ in reward; unclipped unstable. |

✓ = RKL-correct, ✗ = incorrect, ≈ = 1st-order approx. *Full grad required; detaching $\rho \to 1-\delta$. $^\dagger$App. G.

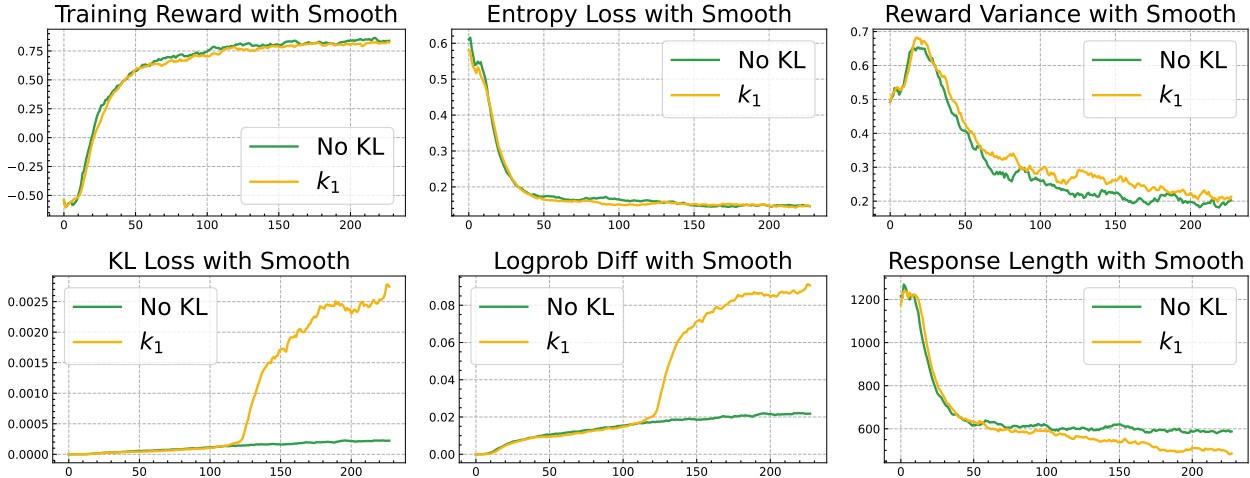

*Figure 2.* **Experiment 1:** 7B-scale comparison of $\boldsymbol{k_1}$ **as loss** versus no KL. As predicted, $\boldsymbol{k_1}$ **as loss** provides no effective regularization; it can increase stochasticity and lead to sharper drift (larger KL and log-prob gaps).

where $\pi_\theta > \pi_{\mathrm{ref}}$, the constraint from $\boldsymbol{k_3}$ **as loss** weakens in later stages. Indeed, both methods achieve comparable reward, but $\boldsymbol{k_2}$ **as loss** maintains smaller log-probability gaps, indicating tighter coupling to the reference. Under a weaker and more realistic constraint ($\beta = 0.001$), $\boldsymbol{k_3}$ **as loss** can show an unstable spike in the KL-loss curve (Figure 4), while $\boldsymbol{k_2}$ **as loss** does not in the same setting.

**Downstream Performance.** Table 3 reports benchmark accuracy after training. Methods without effective KL constraints (no KL, $\boldsymbol{k_1}$ **as loss**) achieve highest scores because they impose no limit on policy drift, while principled regularizers ($\boldsymbol{k_2}$ **as loss**, $\boldsymbol{k_3}$ **as loss**) trade some accuracy for stability—consistent with the intended role of KL regularization. Full results including 1.5B-scale and general reasoning benchmarks appear in Appendix N.

**Summary.** These results corroborate the theoretical analysis: $\boldsymbol{k_1}$ **as loss** provides no regularization signal, while $\boldsymbol{k_2}$ **as loss** enforces a more stable constraint than $\boldsymbol{k_3}$ **as loss**.

*Table 3.* Downstream math accuracy (7B, $\beta = 0.5$). Methods without effective KL regularization allow unconstrained drift, yielding higher scores but less stable training.

| Method | AIME | AMC | MATH | Minerva | Oly. | Avg. |
|---|---|---|---|---|---|---|
| Qwen2.5-Math-7B | 8.2 | 31.3 | 43.6 | 7.4 | 15.6 | 19.0 |
| + RL w/o KL | 17.5 | 55.6 | 78.6 | 36.8 | 42.4 | 41.4 |
| + $k_1$ as loss | 15.4 | 56.0 | 80.6 | 40.8 | 43.0 | 41.8 |
| + $k_2$ as loss | 11.5 | 48.5 | 64.2 | 16.9 | 24.9 | 29.6 |
| + $k_3$ as loss | 13.2 | 48.9 | 65.4 | 18.8 | 29.0 | 31.4 |

**Case Study: DeepSeek-V3.2.** DeepSeek-V3.2 (Liu et al., 2025a) appears to recognize the bias in vanilla $\boldsymbol{k_3}$ **as loss** and proposes a corrected IS-weighted variant:

$$\mathcal{L}_{\mathrm{KL,DS}}(\boldsymbol{\theta}) = \frac{\pi_{\boldsymbol{\theta}}}{\pi_{\mathrm{old}}} \cdot k_3(\boldsymbol{\theta}). \qquad (20)$$

Notably, the ratio $\rho(\boldsymbol{\theta}) = \pi_{\boldsymbol{\theta}}/\pi_{\mathrm{old}}$ is *gradient-carrying*—unlike standard IS where the ratio is detached. Through our framework, we show this design choice is deliberate: the product-rule expansion $\nabla(\rho \cdot k_3)$ generates two terms

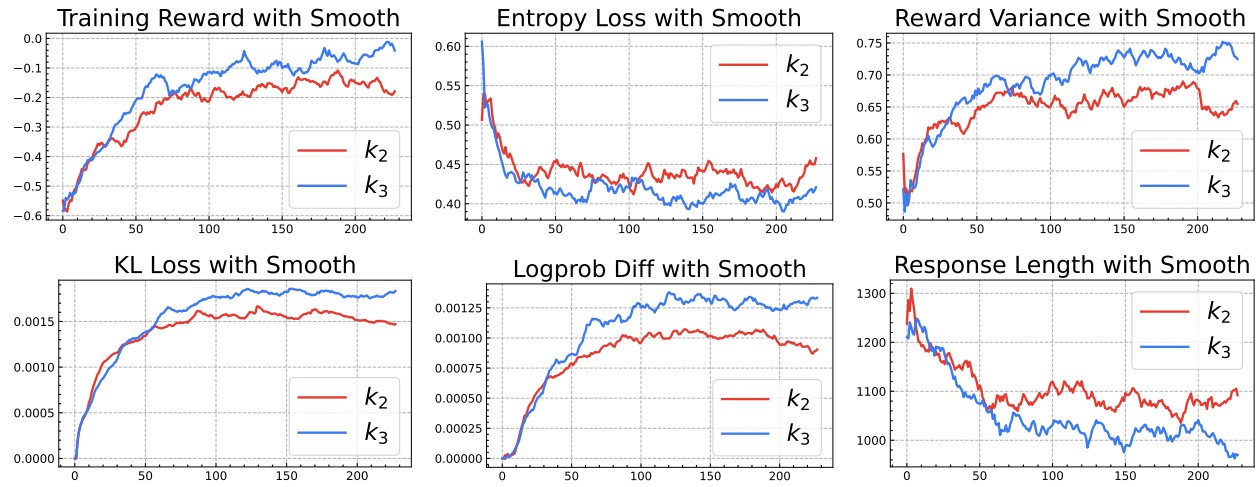

*Figure 3.* **Experiment 2:** 7B-scale comparison of $k_2$ **as loss** (principled RKL) and $k_3$ **as loss** (first-order surrogate). Both constrain the policy, but $k_2$ **as loss** maintains tighter coupling to the reference and yields more stable training dynamics.

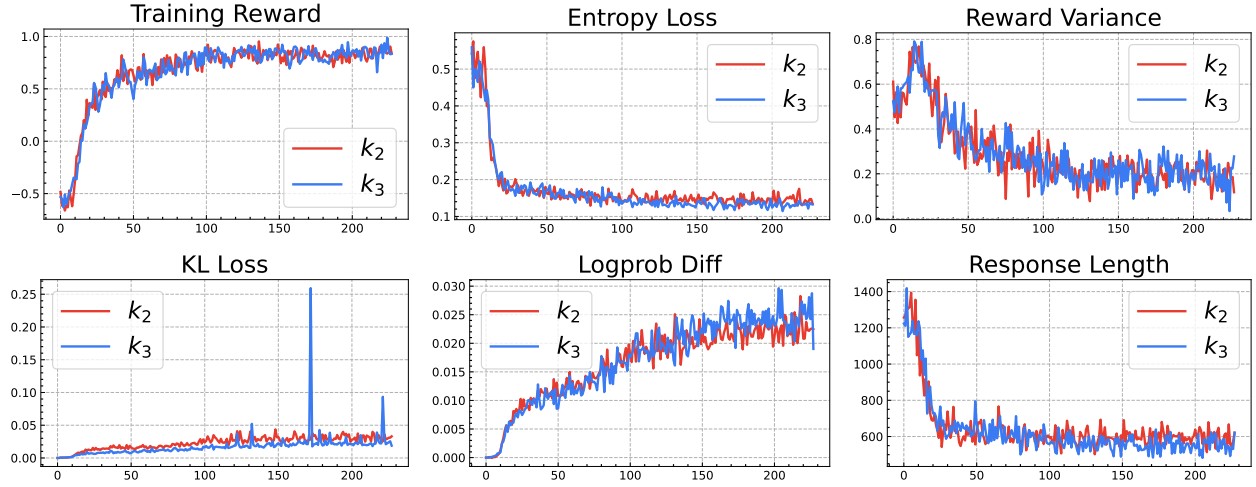

*Figure 4.* 7B-scale training diagnostics for $k_2$ **as loss** and $k_3$ **as loss** at a weaker KL weight ($\beta = 0.001$). While most metrics track closely, the **KL Loss** panel (bottom-left) reveals a visible spike for $k_3$ **as loss** (reaching ∼0.25) that is absent for $k_2$ **as loss**, demonstrating the $k_3$ surrogate's instability when drift is larger. See Figure 7 for 1.5B-scale results showing the same pattern.

whose biases cancel, recovering the principled $k_1$ coefficient. However, this creates a practical tension: preserving the full gradient can yield large updates when $\rho$ is large, while detaching $\rho$ loses the cancellation and reduces to the biased $k_3$ direction. Appendix L provides the full derivation.

## 7. Conclusion

This paper develops a gradient-centric framework for KL regularization in RLHF, expressing each implementation as a scalar coefficient on the policy score function. This perspective yields a simple correctness criterion and three main findings: (i) $k_1$ **as loss** has zero expected gradient despite being an unbiased value estimator; (ii) $k_1$ **in reward** and $k_2$ **as loss** are gradient-equivalent and correctly

implement Reverse KL; (iii) $k_3$ **as loss** is a biased first-order surrogate with mismatched tail behavior. We also identify missing importance-sampling corrections in off-policy "as loss" implementations.

Experiments at 7B scale validate these predictions, with $k_2$ **as loss** showing greater stability than $k_3$ **as loss** under weak regularization. We recommend $k_1$ **in reward** or $k_2$ **as loss** for principled RLHF regularization.

## Impact Statement

This work is methodological and aims to improve the stability and reproducibility of RLHF training by clarifying the effects of KL regularization on gradients. Better-grounded KL implementations can reduce training instability and wasted compute, and may lower the risk of unintended behaviors caused by uncontrolled policy drift during alignment. At the same time, improvements to RLHF optimization can be used in both beneficial and harmful applications; we encourage practitioners to evaluate downstream impacts and follow responsible deployment practices.

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

## Appendix Overview

This appendix provides comprehensive supporting material for the theoretical framework and empirical analysis presented in the main text. To help readers navigate efficiently, we organize the content into four thematic groups.

**I. Foundations and Notation (Section A–Section B).**  These sections establish the practical context and mathematical foundations that underpin our analysis.

- **Section A:** Implementation details and reward shaping in RLHF—how minibatches are structured, how rewards are normalized, and how the KL term is integrated.

- **Section B:** Policy-gradient derivation for reward maximization—clarifying the distinction between objectives, gradients, and surrogate losses.

**II. Core Theoretical Results (Section C–Section E).**  These sections contain the formal proofs supporting the main claims.

- **Section C:** Full proof that $k_1$ in reward and $k_2$ as loss are gradient-equivalent under on-policy sampling—the "gold standard" for RKL regularization.

- **Section D:** Surrogate objectives and Monte Carlo estimation—connecting population-level theory to practical minibatch implementations.

- **Section E:** Formal analysis of the $k_3$ as loss surrogate—proving its first-order bias, tail asymmetry, and variance sensitivity.

**III. Extensions and Alternative Methods (Section F–Section L).**  These sections extend the gradient-centric framework to related settings and alternative designs.

- **Section F:** Derivation of the bounded MiniMax-01 regularizer as an MSE-based alternative with bounded coefficients.

- **Section G:** Off-policy corrections for KL regularization—principled importance sampling and clipping when reusing samples.

- **Section H:** Python code for visualizing the coefficient functions in Figure 1.

- **Section I:** Statistical instability of the $k_3$ value estimator—conditions under which its variance becomes infinite.

- **Section J:** Group normalization stability issues and a proposed fix.

- **Section K:** Forward KL vs. Reverse KL—reinterpreting $k_3$ as loss as a variance-reduced FKL estimator.

- **Section L:** Analysis of the DeepSeek-V3.2 importance-weighted KL estimator and its trade-offs.

**IV. Empirical Supplements (Section M–Section N).**  These sections provide additional experimental evidence.

- **Section M:** 1.5B-scale experiments complementing the 7B-scale results in the main text.

- **Section N:** Downstream benchmark performance on math and general reasoning tasks.

## A. Detailed Implementation of RLHF Methods

The main text introduces a unified framework that analyzes KL implementations through their induced gradient coefficients. Before diving into the theoretical analysis, it is helpful to establish the practical context: how are RLHF algorithms actually implemented, and where do the different KL formulations fit in?

This section provides that context. We describe the standard minibatch structure, the reward shaping techniques used in practice, and the two canonical ways to integrate KL regularization—the "combined form" ($k_n$ in reward) and the "decoupled form" ($k_n$ as loss). These details will be referenced throughout the theoretical sections that follow.

**Minibatch structure.** In a typical training step, we draw $N$ prompts $\{x^{(i)}\}_{i=1}^N$ from the dataset $\mathcal{D}$. For each prompt $x^{(i)}$, we sample $G$ responses $y^{(i,1)}, \ldots, y^{(i,G)} \sim \pi_\theta(\cdot \mid x^{(i)})$ from the detached policy snapshot. The $G$ responses for each prompt form a *group*; the full minibatch contains $N \times G$ prompt–response pairs. We use $\pi_{\boldsymbol{\theta}}$ for the trainable policy (gradients flow through it) and $\pi_\theta$ for its detached snapshot.

**Baseline subtraction and normalization.** Having established the minibatch structure, we now describe how the raw reward signal is processed before being used in the policy gradient. A key technique is baseline subtraction: subtracting an action-independent baseline does not change the expected policy gradient (see Equation (36)) and typically reduces variance (Williams, 1992). We consider the following operators applied to a scalar signal $r$:

$$f_{\text{group-bl}}(r) = r - \text{mean}_{\text{group}}(r), \tag{21}$$

$$f_{\text{batch-bl}}(r) = r - \text{mean}_{\text{batch}}(r). \tag{22}$$

In practice, normalization is also used. Normalization is *not* unbiased but can improve numerical stability by controlling the scale of the reward signal:

$$f_{\text{BN}}(r) = \frac{r - \text{mean}_{\text{batch}}(r)}{\text{std}_{\text{batch}}(r)}, \tag{23}$$

$$f_{\text{GN}}(r) = \frac{r - \text{mean}_{\text{group}}(r)}{\text{std}_{\text{group}}(r)}. \tag{24}$$

**How common algorithms shape the reward signal.** With these operators defined, we can now specify how different RLHF algorithms process the raw reward signal. Let $r_{\text{raw}}(x,y)$ denote the raw score from the reward model. Different algorithms post-process this score as follows:

$$\textbf{REINFORCE:} \quad r(x,y) = f_{\text{BN}}\big(r_{\text{raw}}(x,y)\big), \tag{25}$$

$$\textbf{PPO/REINFORCE++:} \quad r(x,y) = f_{\text{BN}}\big(r_{\text{raw}}(x,y)\big), \tag{26}$$

$$\textbf{GRPO:} \quad r(x,y) = f_{\text{GN}}\big(r_{\text{raw}}(x,y)\big), \tag{27}$$

$$\textbf{Dr-GRPO (Liu et al., 2025c):} \quad r(x,y) = f_{\text{group-bl}}\big(r_{\text{raw}}(x,y)\big), \tag{28}$$

$$\textbf{REINFORCE++baseline:} \quad r(x,y) = f_{\text{BN}}\Big(f_{\text{group-bl}}\big(r_{\text{raw}}(x,y)\big)\Big). \tag{29}$$

**Integration of the KL regularization loss.** The transforms above shape only the reward signal. A separate and critical design choice is how to integrate the KL regularizer. As discussed in the main text (Section 4), there are two canonical approaches:

(i) **Combined form ($k_n$ in reward):** Used by REINFORCE/PPO methods. A combined reward signal is formed first,

$$A_{\text{combined}}(x,y) \;=\; r_{\text{raw}}(x,y) \;-\; \beta\, k_1\Big(\pi_\theta(y \mid x),\, \pi_{\text{ref}}(y \mid x)\Big),$$

and then baseline/normalization is applied to this combined signal $A_{\text{combined}}$ before it multiplies the score function.

(ii) **Decoupled form ($k_n$ as loss):** Used by GRPO methods. The KL penalty $k_n\big(\pi_{\boldsymbol{\theta}}(\cdot \mid x), \pi_{\text{ref}}(\cdot \mid x)\big)$ is optimized as a separate, unnormalized loss term, added to the policy-gradient loss driven by the shaped reward $r(x,y)$.

**Complete on-policy objectives.** Combining the reward shaping and KL integration described above, we can now write down the complete on-policy objectives for representative algorithms. These expressions make explicit how the two KL styles manifest in practice.

**REINFORCE/PPO (Monte Carlo minibatch):**

$$\mathcal{L}_{\text{REINFORCE/PPO,MC}}(\boldsymbol{\theta}) = -\frac{1}{NG} \sum_{i=1}^N \sum_{j=1}^G \Big\{ A\big(x^{(i)}, y^{(i,j)}\big) \log \pi_{\boldsymbol{\theta}}\big(y^{(i,j)} \mid x^{(i)}\big) \Big\}, \tag{30}$$

$$\text{where } A(x,y) = f_{\text{BN}}\Big(r_{\text{raw}}(x,y) - \beta\, k_1\Big(\pi_\theta(y \mid x),\, \pi_{\text{ref}}(y \mid x)\Big)\Big). \tag{31}$$

Here, the $k_1$ term is evaluated at the current policy snapshot (no gradients flow through it), consistent with the policy-gradient framework where it acts as a coefficient for the score function.

**GRPO (Monte Carlo minibatch):**

$$\mathcal{L}_{\text{GRPO,MC}}(\boldsymbol{\theta}) = -\frac{1}{NG} \sum_{i=1}^{N} \sum_{j=1}^{G} \left\{ r\big(x^{(i)}, y^{(i,j)}\big) \log \pi_{\boldsymbol{\theta}}\big(y^{(i,j)} \mid x^{(i)}\big) \right\}$$
$$+ \frac{\beta}{NG} \sum_{i=1}^{N} \sum_{j=1}^{G} k_3\Big(\pi_{\boldsymbol{\theta}}\big(y^{(i,j)} \mid x^{(i)}\big),\ \pi_{\text{ref}}\big(y^{(i,j)} \mid x^{(i)}\big)\Big). \tag{32}$$

In this decoupled form, the shaped reward $r(x, y)$ drives the policy-gradient term, while the KL penalty is a separate loss where gradients flow directly through $\pi_{\boldsymbol{\theta}}$ inside $k_3(\cdot)$.

*Remark.* While baseline subtraction is an unbiased variance-reduction technique, normalization is a biased but often crucial heuristic for practical stability. Both are important engineering details, even if omitted from simplified theoretical analyses.

# B. Policy Gradient Derivation for Reward Maximization

The gradient-centric framework developed in the main text relies on expressing any KL implementation as a coefficient multiplying the policy score function $\nabla_{\boldsymbol{\theta}} \log \pi_{\boldsymbol{\theta}}(y|x)$. Before analyzing KL regularization specifically, it is instructive to review the standard policy gradient derivation for reward maximization, as the same techniques—the log-derivative trick and the distinction between trainable and detached parameters—apply throughout.

This section derives the policy gradient for the reward-maximization objective, clarifying the distinction between the true objective, its gradient, and surrogate loss functions. We adopt the same notation convention as the main text: $\boldsymbol{\theta}$ denotes parameters that carry gradients, while $\theta$ denotes detached parameters (e.g., in the sampling distribution).

**Objective.** The goal is to maximize expected reward:

$$\mathcal{J}_{\text{reward}}(\boldsymbol{\theta}) = \mathbb{E}_{x \sim \mathcal{D}, y \sim \pi_{\boldsymbol{\theta}}(\cdot|x)} \left[ r(x, y) \right] = \mathbb{E}_{x \sim \mathcal{D}} \sum_{y} \left[ r(x, y) \cdot \pi_{\boldsymbol{\theta}}(y|x) \right]. \tag{33}$$

We assume standard regularity conditions that permit the interchange of differentiation and expectation operators.

**Policy gradient derivation.** We compute the gradient of the objective function $\mathcal{J}_{\text{reward}}(\boldsymbol{\theta})$ using the log-derivative trick. The distinction between $\boldsymbol{\theta}$ (trainable) and $\theta$ (detached) is crucial: the former appears wherever gradients must flow, while the latter marks quantities that are held fixed during differentiation.

$$\nabla_{\boldsymbol{\theta}} \mathcal{J}_{\text{reward}}(\boldsymbol{\theta}) = \nabla_{\boldsymbol{\theta}} \mathbb{E}_{x \sim \mathcal{D}} \sum_{y} \left[ r(x, y) \cdot \pi_{\boldsymbol{\theta}}(y|x) \right]$$
$$= \mathbb{E}_{x \sim \mathcal{D}} \sum_{y} \left[ r(x, y) \cdot \nabla_{\boldsymbol{\theta}} \pi_{\boldsymbol{\theta}}(y|x) \right]$$
$$= \mathbb{E}_{x \sim \mathcal{D}} \sum_{y} \Big[ r(x, y) \cdot \pi_{\theta}(y|x) \cdot$$
$$\frac{\nabla_{\boldsymbol{\theta}} \pi_{\boldsymbol{\theta}}(y|x)}{\pi_{\theta}(y|x)} \Big] \tag{34}$$
$$= \mathbb{E}_{x \sim \mathcal{D}} \sum_{y} \pi_{\theta}(y|x) \left[ r(x, y) \cdot \nabla_{\boldsymbol{\theta}} \log \pi_{\boldsymbol{\theta}}(y|x) \right]$$
$$= \mathbb{E}_{x \sim \mathcal{D}, y \sim \pi_{\theta}(\cdot|x)} \left[ r(x, y) \cdot \nabla_{\boldsymbol{\theta}} \log \pi_{\boldsymbol{\theta}}(y|x) \right].$$

The key step occurs in the third line, where we multiply and divide by $\pi_{\theta}(y|x)$. This serves two purposes: (i) it converts the sum over $y$ into an expectation under $\pi_{\theta}$, enabling Monte Carlo estimation; and (ii) it introduces the log-derivative identity $\nabla_{\boldsymbol{\theta}} \log \pi_{\boldsymbol{\theta}} = \nabla_{\boldsymbol{\theta}} \pi_{\boldsymbol{\theta}} / \pi_{\theta}$. In the final line, the gradient is expressed as an expectation over samples from $\pi_{\theta}$, where the sampling process itself has no gradient path—only the score function $\nabla_{\boldsymbol{\theta}} \log \pi_{\boldsymbol{\theta}}$ carries gradients. This structure, where a scalar coefficient (without gradients) multiplies the score function, is the foundation for the unified framework in Section 4.

# C. Formal Proof of the Principled KL Regularization

This section provides the complete proof of Theorem 5.1 from the main text—the central theoretical result establishing that $k_1$ **in reward** and $k_2$ **as loss** are gradient-equivalent implementations of the Reverse KL (RKL) regularizer.

The proof proceeds in four stages. First, we state the regularity assumptions required for the analysis (Section C.1). Second, we establish two fundamental identities—the log-derivative identity and the zero-mean score property—that underpin all policy-gradient derivations (Section C.2). Third, we derive the exact gradient of the true RKL objective (Section C.3). Finally, we show that both surrogates reproduce this target gradient (Section C.4).

Throughout, we use $\pi_{\boldsymbol{\theta}}(\cdot|x)$ for the gradient-carrying policy and $\pi_\theta(\cdot|x)$ for its detached snapshot (numerically identical at the current iterate). Sampling measures and scalar coefficients multiplying the score function are always treated as detached.

## C.1. Assumptions and Notation

We begin by stating the regularity assumptions that ensure all subsequent derivations are well-defined. Let $\mathcal{D}$ be a data distribution over prompts $x$, $\pi_{\text{ref}}(\cdot|x)$ a fixed reference policy, and $\pi_{\boldsymbol{\theta}}(\cdot|x)$ a differentiable policy parameterized by $\boldsymbol{\theta}$. All logarithms are natural.

**(A1)** **(Valid policy)** For each $x$, the function $y \mapsto \pi_{\boldsymbol{\theta}}(y|x)$ is a valid probability mass/density: $\pi_{\boldsymbol{\theta}}(y|x) > 0$ on its support, it is differentiable in $\boldsymbol{\theta}$, and normalizes to one, i.e., $\sum_y \pi_{\boldsymbol{\theta}}(y|x) = 1$ (or $\int \pi_{\boldsymbol{\theta}}(y|x)\, dy = 1$).

**(A2)** **(Interchange)** The interchange of expectation/summation and differentiation is valid (standard regularity conditions).

**(A3)** **(Independence)** The data distribution $\mathcal{D}$ and reference policy $\pi_{\text{ref}}$ do not depend on the trainable parameters $\boldsymbol{\theta}$.

**(A4)** **(Support)** The KL divergence is well-defined: for all $x$ and all $y$ in the support of $\pi_{\boldsymbol{\theta}}(\cdot|x)$, we have $\pi_{\text{ref}}(y|x) > 0$.

**Notation convention**: Throughout this section, $\pi_\theta$ (black $\theta$) denotes the *detached* copy of $\pi_{\boldsymbol{\theta}}$ (red $\boldsymbol{\theta}$), numerically equal at the current iterate but with no gradient path. Unless stated otherwise, expectations over $y$ are taken with respect to the detached sampling distribution $y \sim \pi_\theta(\cdot|x)$.

## C.2. Fundamental Identities

Before deriving the RKL gradient, we establish two lemmas and two corollaries that will be used repeatedly. These identities formalize the key properties of the score function that make policy-gradient methods tractable.

**Lemma C.1** (Log-derivative identity with detached denominator). *For any fixed $x$ and any $y$ with $\pi_\theta(y|x) > 0$,*

$$\nabla_{\boldsymbol{\theta}} \log \pi_{\boldsymbol{\theta}}(y|x) = \frac{\nabla_{\boldsymbol{\theta}} \pi_{\boldsymbol{\theta}}(y|x)}{\pi_\theta(y|x)}. \tag{35}$$

*Proof.* By the chain rule, $\nabla_\theta \log \pi_\theta = (\nabla_\theta \pi_\theta)/\pi_\theta$. Replacing the denominator with its detached, numerically identical copy $\pi_\theta$ preserves the numerical value while making the no-gradient path explicit. □ □

**Lemma C.2** (Zero-mean score). *For any fixed $x$,*

$$\mathbb{E}_{y \sim \pi_\theta(\cdot|x)}\big[\nabla_{\boldsymbol{\theta}} \log \pi_{\boldsymbol{\theta}}(y|x)\big] = 0. \tag{36}$$

*Proof.* Using Lemma C.1,

$$\sum_y \pi_\theta(y|x)\, \frac{\nabla_{\boldsymbol{\theta}} \pi_{\boldsymbol{\theta}}(y|x)}{\pi_\theta(y|x)} = \sum_y \nabla_{\boldsymbol{\theta}} \pi_{\boldsymbol{\theta}}(y|x) = \nabla_{\boldsymbol{\theta}} \sum_y \pi_{\boldsymbol{\theta}}(y|x) = \nabla_{\boldsymbol{\theta}}(1) = 0. \tag{37}$$

□

**Corollary C.0.1** (Score-function reweighting). *For any function $z(y, x)$ that does not depend on $\boldsymbol{\theta}$,*

$$\sum_y \nabla_{\boldsymbol{\theta}} \pi_{\boldsymbol{\theta}}(y|x)\, z(y, x) = \mathbb{E}_{y \sim \pi_\theta(\cdot|x)}\big[z(y, x)\, \nabla_{\boldsymbol{\theta}} \log \pi_{\boldsymbol{\theta}}(y|x)\big]. \tag{38}$$

*Proof.* Multiply and divide by $\pi_\theta(y|x)$, then apply Lemma C.1:

$$\sum_y \nabla_{\boldsymbol{\theta}} \pi_\theta(y|x)\, z(y,x) = \sum_y \pi_\theta(y|x)\, \frac{\nabla_{\boldsymbol{\theta}} \pi_\theta(y|x)}{\pi_\theta(y|x)}\, z(y,x) = \sum_y \pi_\theta(y|x)\, z(y,x)\, \nabla_{\boldsymbol{\theta}} \log \pi_\theta(y|x).$$

The right-hand side is precisely the expectation $\mathbb{E}_{y\sim\pi_\theta(\cdot|x)}[z(y,x)\,\nabla_{\boldsymbol{\theta}} \log \pi_\theta(y|x)]$. $\square$

**Corollary C.0.2** (Baseline invariance)**.** *For any function $b(x)$ that does not depend on $\boldsymbol{\theta}$,*

$$\mathbb{E}_{y\sim\pi_\theta(\cdot|x)}\big[b(x)\,\nabla_{\boldsymbol{\theta}} \log \pi_\theta(y|x)\big] = b(x) \cdot \mathbb{E}_{y\sim\pi_\theta(\cdot|x)}\big[\nabla_{\boldsymbol{\theta}} \log \pi_\theta(y|x)\big] = 0. \tag{39}$$

*Thus, adding an action-independent baseline $b(x)$ (which does not carry gradients) to any coefficient does not change the expected gradient.*

### C.3. Derivation of the True RKL Gradient

With the fundamental identities in hand, we now derive the gradient of the true RKL objective. This gradient serves as the "target" against which we will compare the two surrogate formulations. The RKL divergence objective is:

$$\mathcal{J}_{\text{RKL}}(\boldsymbol{\theta}) = \mathbb{E}_{x\sim\mathcal{D}} \left[ \sum_y \pi_\theta(y|x)\, \log \frac{\pi_\theta(y|x)}{\pi_{\text{ref}}(y|x)} \right]. \tag{40}$$

**Step 1 (Differentiate under the expectation).** By (A2), the gradient operator is moved inside the expectation and sum.

**Step 2 (Apply product rule).** For each $y$, we differentiate the term $\pi_\theta(y|x)\log \pi_\theta(y|x)$ using the product rule. Let $f = \pi_\theta$ and $g = \log \pi_\theta$. Then:

$$\begin{aligned} \nabla_{\boldsymbol{\theta}}\big[\pi_\theta \log \pi_\theta\big] &= (\nabla_{\boldsymbol{\theta}} \pi_\theta)\, \log \pi_\theta + \pi_\theta \cdot \nabla_{\boldsymbol{\theta}} \log \pi_\theta \\ &= (\nabla_{\boldsymbol{\theta}} \pi_\theta)\, \log \pi_\theta + \pi_\theta \cdot \frac{\nabla_{\boldsymbol{\theta}} \pi_\theta}{\pi_\theta} \quad \text{(by Lemma C.1)} \\ &= (\nabla_{\boldsymbol{\theta}} \pi_\theta)\, (\log \pi_\theta + 1). \end{aligned} \tag{41}$$

Here, the factors $\log \pi_\theta$ and $\pi_\theta$ that are not being differentiated are written as their detached copies to emphasize the no-gradient path. For the reference term, by (A3) $\pi_{\text{ref}}$ does not depend on $\boldsymbol{\theta}$, so:

$$\nabla_{\boldsymbol{\theta}}\big[-\pi_\theta(y|x) \log \pi_{\text{ref}}(y|x)\big] = -(\nabla_{\boldsymbol{\theta}} \pi_\theta(y|x))\, \log \pi_{\text{ref}}(y|x). \tag{42}$$

**Step 3 (Collect terms).** Combining the two terms from Step 2, we obtain:

$$\nabla_{\boldsymbol{\theta}} \mathcal{J}_{\text{RKL}}(\boldsymbol{\theta}) = \mathbb{E}_{x\sim\mathcal{D}} \left[ \sum_y \nabla_{\boldsymbol{\theta}} \pi_\theta(y|x)\, \left( \log \frac{\pi_\theta(y|x)}{\pi_{\text{ref}}(y|x)} + 1 \right) \right]. \tag{43}$$

At this stage, the gradient is expressed as a sum over $y$, weighted by $\nabla_{\boldsymbol{\theta}} \pi_\theta(y|x)$. To convert this into a form amenable to Monte Carlo estimation, we apply Corollary C.0.1.

**Step 4 (Apply score-function reweighting).** Setting the scalar coefficient $z(y,x) := \log \frac{\pi_\theta(y|x)}{\pi_{\text{ref}}(y|x)} + 1$ (which does not carry gradients) and applying Corollary C.0.1:

$$\nabla_{\boldsymbol{\theta}} \mathcal{J}_{\text{RKL}}(\boldsymbol{\theta}) = \mathbb{E}_{x\sim\mathcal{D},\, y\sim\pi_\theta(\cdot|x)} \left[ \left( \log \frac{\pi_\theta(y|x)}{\pi_{\text{ref}}(y|x)} + 1 \right) \nabla_{\boldsymbol{\theta}} \log \pi_\theta(y|x) \right]. \tag{44}$$

**Step 5 (Simplify using the zero-mean score property).** The coefficient in Equation (44) contains a constant $+1$ term. By Lemma C.2, this term contributes zero to the expected gradient:

$$\mathbb{E}_{y\sim\pi_\theta(\cdot|x)} \left[ 1 \cdot \nabla_{\boldsymbol{\theta}} \log \pi_\theta(y|x) \right] = 0.$$

Dropping this zero-contribution term yields the final, simplified gradient:

$$\nabla_{\boldsymbol{\theta}} \mathcal{J}_{\text{RKL}}(\boldsymbol{\theta}) = \mathbb{E}_{x\sim\mathcal{D},\, y\sim\pi_\theta(\cdot|x)} \left[ \log \frac{\pi_\theta(y|x)}{\pi_{\text{ref}}(y|x)}\, \nabla_{\boldsymbol{\theta}} \log \pi_\theta(y|x) \right]. \tag{45}$$

**C.4. The Gold Standard: $k_1$ in reward $\Leftrightarrow k_2$ as loss**

Having derived the true RKL gradient in Equation (45), we now show that two structurally different surrogates—$k_1$ **in reward** and $k_2$ **as loss**—both reproduce this exact gradient under on-policy sampling. This equivalence is the core theoretical result that validates both as principled implementations of RKL regularization.

**Surrogate 1: $k_1$ in reward.** The first surrogate treats the log-ratio as a scalar coefficient multiplying the score function:

$$\mathcal{J}_{k_1 \text{ in reward}}(\boldsymbol{\theta}) = \mathbb{E}_{x \sim \mathcal{D}, \, y \sim \pi_\theta(\cdot|x)} \Big[ \underbrace{\Big( \log \frac{\pi_\theta(y|x)}{\pi_{\text{ref}}(y|x)} \Big)}_{\text{coefficient } k_1} \log \pi_{\boldsymbol{\theta}}(y|x) \Big]. \tag{46}$$

Since the coefficient $k_1$ does not carry gradients (it is evaluated at the current snapshot $\pi_\theta$), differentiating yields:

$$\nabla_{\boldsymbol{\theta}} \mathcal{J}_{k_1 \text{ in reward}}(\boldsymbol{\theta}) = \mathbb{E}_{x \sim \mathcal{D}, \, y \sim \pi_\theta(\cdot|x)} \Big[ \Big( \log \frac{\pi_\theta(y|x)}{\pi_{\text{ref}}(y|x)} \Big) \nabla_{\boldsymbol{\theta}} \log \pi_{\boldsymbol{\theta}}(y|x) \Big], \tag{47}$$

which is identical to Equation (45). This is the style used in PPO.

**Surrogate 2: $k_2$ as loss.** The second surrogate differentiates *through* the log-ratio via a squared penalty:

$$\mathcal{J}_{k_2 \text{ as loss}}(\boldsymbol{\theta}) = \mathbb{E}_{x \sim \mathcal{D}, \, y \sim \pi_\theta(\cdot|x)} \Big[ \frac{1}{2} \big( \log \pi_{\boldsymbol{\theta}}(y|x) - \log \pi_{\text{ref}}(y|x) \big)^2 \Big]. \tag{48}$$

Applying the chain rule to differentiate the squared term:

$$\nabla_{\boldsymbol{\theta}} \Big[ \frac{1}{2} \big( \log \pi_{\boldsymbol{\theta}} - \log \pi_{\text{ref}} \big)^2 \Big] = \big( \log \pi_{\boldsymbol{\theta}} - \log \pi_{\text{ref}} \big) \cdot \nabla_{\boldsymbol{\theta}} \log \pi_{\boldsymbol{\theta}}. \tag{49}$$

Evaluating the scalar multiplier at the current policy snapshot $\pi_\theta$ and taking the expectation:

$$\nabla_{\boldsymbol{\theta}} \mathcal{J}_{k_2 \text{ as loss}}(\boldsymbol{\theta}) = \mathbb{E}_{x \sim \mathcal{D}, \, y \sim \pi_\theta(\cdot|x)} \Big[ \big( \log \pi_\theta(y|x) - \log \pi_{\text{ref}}(y|x) \big) \nabla_{\boldsymbol{\theta}} \log \pi_{\boldsymbol{\theta}}(y|x) \Big], \tag{50}$$

which is also identical to Equation (45). Thus, despite their different functional forms, both surrogates induce the same expected gradient—the principled RKL gradient.

**C.5. Conclusion and Implementation Guidance**

We have now established the central result: under Assumptions (A1)–(A4), the true RKL objective and both surrogates—$k_1$ **in reward** and $k_2$ **as loss**—share the same expected gradient, given by Equation (45). This equivalence rests on the following key conventions, which practitioners should adhere to for correct implementation.

**Sampling Measure**: Samples are drawn from the current policy snapshot $y \sim \pi_\theta(\cdot|x)$ (e.g., via an external rollout engine).

**Scalar Coefficients**: The scale coefficients that multiply the score function do not carry gradients—they are evaluated at the current snapshot $\pi_\theta$. Applying Corollary C.0.2, any action-independent baseline $b(x)$ can be added to reduce variance.

**Gradient Path**: Gradients propagate only through terms explicitly parameterized by $\boldsymbol{\theta}$.

**Implementation Notes.** We now translate these theoretical results into concrete implementation patterns. For a single on-policy sample $y \sim \pi_\theta(\cdot|x)$:

- $k_1$ **in reward**: The surrogate loss is

$$\mathcal{L}_{k_1 \text{ in reward}} = \big( \log \pi_\theta(y|x) - \log \pi_{\text{ref}}(y|x) \big) \cdot \log \pi_{\boldsymbol{\theta}}(y|x),$$

where the coefficient $k_1 = \log \pi_\theta - \log \pi_{\text{ref}}$ is evaluated at the current snapshot (stop gradient) and only $\log \pi_{\boldsymbol{\theta}}$ carries gradients. Gradient descent on $\mathcal{L}_{k_1 \text{ in reward}}$ performs descent on $\mathcal{J}_{\text{RKL}}$. In typical RLHF, where we maximize $\mathcal{J}_{\text{Reward}} - \beta \, \mathcal{J}_{\text{RKL}}$, the combined loss to minimize is $-\big( r(x,y) - \beta \, k_1 \big) \cdot \log \pi_{\boldsymbol{\theta}}(y|x)$.

- **$k_2$ as loss**: The surrogate loss is

$$\mathcal{L}_{k_2 \text{ as loss}} = \tfrac{1}{2}\big(\log \pi_{\boldsymbol{\theta}}(y|x) - \log \pi_{\text{ref}}(y|x)\big)^2,$$

where the entire log-ratio carries gradients through $\pi_{\boldsymbol{\theta}}$. Gradient descent on $\mathcal{L}_{k_2 \text{ as loss}}$ also performs descent on $\mathcal{J}_{\text{RKL}}$, since differentiating yields the same coefficient $k_1 = \log \pi_\theta - \log \pi_{\text{ref}}$ multiplying $\nabla_{\boldsymbol{\theta}} \log \pi_{\boldsymbol{\theta}}$.

**Remarks.** (i) For continuous spaces, replace sums by integrals; the proof is unchanged provided densities are positive on their support. (ii) The equivalence requires on-policy sampling. If samples are drawn from a stale policy $\pi_{\text{old}}$, exact correction uses importance weights $\rho(x, y) = \pi_\theta(y|x)/\pi_{\text{old}}(y|x)$ inside the expectations.

## D. Surrogate Objective: Full-Vocabulary vs. Monte Carlo

The preceding sections derive policy gradients as expectations over the full action space. In practice, however, we cannot enumerate all possible responses $y$ for a given prompt $x$—the vocabulary is astronomically large for language models. This section bridges the gap between theory and practice by formalizing the connection between population-level objectives and their minibatch Monte Carlo estimators.

We detail two conceptually distinct implementations: a **full-vocabulary loss**, which computes the exact inner expectation over all actions (theoretically exact but computationally infeasible), and a **Monte Carlo (MC) loss**, which replaces this sum with i.i.d. samples (unbiased and practical). Understanding this connection is essential for verifying that practical implementations are gradient-correct.

**Conventions and gradient paths.** To ensure consistency with the main text, we restate the key conventions governing gradient flow:

1. The trainable policy $\pi_{\boldsymbol{\theta}}$ carries gradients; its numerically identical snapshot at the current iterate (which does not carry gradients) is denoted $\pi_\theta$.

2. All scalars that multiply the score function do not carry gradients: the reward $r(x, y)$, any KL-derived term $k_n(\cdot)$, and their combination $r(x, y) - \beta\, k_n(\cdot)$.

3. Gradients flow only through $\log \pi_{\boldsymbol{\theta}}(y|x)$; everything inside the coefficient $c(x, y)$ does not carry gradients.

4. We adopt the naming from the main text: "reward coefficient," "$k_n$ coefficient," and "combined form coefficient."

We express the objective using a generic scalar coefficient $c(x, y)$, which can take several forms:

$$c(x, y) \in \left\{ \begin{array}{ll} r(x, y) & \text{reward coefficient} \\ k_n\big(\pi_\theta(y|x),\, \pi_{\text{ref}}(y|x)\big) & k_n \text{ coefficient} \\ r(x, y) - \beta\, k_n\big(\pi_\theta(y|x),\, \pi_{\text{ref}}(y|x)\big) & \text{combined form coefficient} \end{array} \right\}. \tag{51}$$

A typical KL choice is

$$k_1(y|x) = \log \pi_\theta(y|x) - \log \pi_{\text{ref}}(y|x). \tag{52}$$

**Policy gradient in expectation form (with baseline).** For the population objective $\mathcal{J}_{\text{true}}(\boldsymbol{\theta}) = \mathbb{E}_{x \sim \mathcal{D},\, y \sim \pi_{\boldsymbol{\theta}}(\cdot|x)}\big[\, c(x, y)\,\big]$, using an action-independent baseline $b(x)$ and the log-derivative identity (where the denominator is evaluated at the current snapshot),

$$\nabla_{\boldsymbol{\theta}} \log \pi_{\boldsymbol{\theta}}(y|x) = \frac{\nabla_{\boldsymbol{\theta}} \pi_{\boldsymbol{\theta}}(y|x)}{\pi_\theta(y|x)}, \tag{53}$$

the unbiased policy gradient is

$$\nabla_{\boldsymbol{\theta}} \mathcal{J}_{\text{true}}(\boldsymbol{\theta}) = \mathbb{E}_{x \sim \mathcal{D},\, y \sim \pi_\theta(\cdot|x)}\Big[\big(c(x, y) - b(x)\big)\, \nabla_{\boldsymbol{\theta}} \log \pi_{\boldsymbol{\theta}}(y|x)\Big]. \tag{54}$$

This relies on the zero-mean score property under $y \sim \pi_\theta(\cdot|x)$ as proved in Equation (36), ensuring $b(x)$ does not change the expected gradient.

**Population surrogate loss.** A surrogate loss whose negative gradient recovers Equation (54) is

$$\mathcal{L}_{\mathrm{sur}}(\boldsymbol{\theta}) = - \mathbb{E}_{x\sim\mathcal{D},\,y\sim\pi_\theta(\cdot|x)}\Big[\big(c(x,y)-b(x)\big)\,\log\pi_{\boldsymbol{\theta}}(y|x)\Big]. \tag{55}$$

**Two interchangeable minibatch implementations.** We now provide two equivalent minibatch estimators of Equation (55). The first computes the exact inner expectation over the discrete action space $\mathcal{V}$ by summing all actions with a sampling weight—this is the "full-vocabulary" version. The second replaces this inner sum with i.i.d. on-policy samples, yielding an unbiased estimate conditional on the minibatch prompts—this is the "Monte Carlo" version used in practice.

$$\mathcal{L}_{\mathrm{sur,Full}}(\boldsymbol{\theta}) = -\frac{1}{N}\sum_{i=1}^{N}\sum_{y^{(i)}\in\mathcal{V}}\underbrace{\pi_\theta\big(y^{(i)}\mid x^{(i)}\big)}_{\text{sampling weight}}\Big(c\big(x^{(i)},y^{(i)}\big)-b(x^{(i)})\Big)\log\pi_{\boldsymbol{\theta}}\big(y^{(i)}\mid x^{(i)}\big). \tag{56}$$

Here, the sampling weight $\pi_\theta$ does not carry gradients; the gradient path is solely via $\log\pi_{\boldsymbol{\theta}}$.

$$\mathcal{L}_{\mathrm{sur,MC}}(\boldsymbol{\theta}) = -\frac{1}{N}\sum_{i=1}^{N}\frac{1}{G}\sum_{j=1}^{G}\Big(c\big(x^{(i)},y^{(i,j)}\big)-b(x^{(i)})\Big)\log\pi_{\boldsymbol{\theta}}\big(y^{(i,j)}\mid x^{(i)}\big),\quad y^{(i,j)}\sim\pi_\theta(\cdot|x^{(i)}). \tag{57}$$

In Equation (57), $\{y^{(i,j)}\}_{j=1}^{G}$ are i.i.d. samples from the current policy snapshot $\pi_\theta(\cdot|x^{(i)})$; increasing $G$ reduces variance while preserving unbiasedness.

**Unbiasedness and practical considerations.** For any fixed $x^{(i)}$ and function $f$,

$$\mathbb{E}_{\{y^{(i,j)}\}_{j=1}^{G}\text{ i.i.d.}\sim\pi_\theta(\cdot|x^{(i)})}\left[\frac{1}{G}\sum_{j=1}^{G}f\big(y^{(i,j)}\big)\right] = \sum_{y^{(i)}\in\mathcal{V}}\pi_\theta\big(y^{(i)}\mid x^{(i)}\big)\,f\big(y^{(i)}\big), \tag{58}$$

hence $\mathbb{E}\big[\mathcal{L}_{\mathrm{sur,MC}}\mid\{x^{(i)}\}\big]=\mathcal{L}_{\mathrm{sur,Full}}$. In practice, computing $r(x,y)$ or $k_n(\cdot)$ over the full vocabulary is infeasible for LLMs due to GPU memory constraints; MC estimation is therefore standard.

**Alternative decoupled formulation: $k_n$ as loss.** In addition to incorporating KL via the coefficient $c(x,y)$, one may add a separate penalty-only loss that differentiates directly through the log-ratio. Let $\psi_n:\mathbb{R}\to\mathbb{R}$ be differentiable (e.g., $\psi_1(t)=t$, $\psi_2(t)=\frac{1}{2}t^2$). Define

$$\mathcal{L}_{k_n\text{ as loss,MC}}(\boldsymbol{\theta}) = -\frac{1}{NG}\sum_{i=1}^{N}\sum_{j=1}^{G}\psi_n\Big(\underbrace{\log\pi_{\boldsymbol{\theta}}(y^{(i,j)}|x^{(i)})}_{\text{with grad}}-\underbrace{\log\pi_{\mathrm{ref}}(y^{(i,j)}|x^{(i)})}_{\text{fixed}}\Big). \tag{59}$$

Its gradient takes the score-like form

$$\nabla_{\boldsymbol{\theta}}\mathcal{L}_{k_n\text{ as loss,MC}}(\boldsymbol{\theta}) = -\frac{1}{NG}\sum_{i=1}^{N}\sum_{j=1}^{G}\psi_n'(\cdot)\,\nabla_{\boldsymbol{\theta}}\log\pi_{\boldsymbol{\theta}}(y^{(i,j)}|x^{(i)}), \tag{60}$$

where $(\cdot)$ denotes the log-ratio in Equation (59). Here the prime denotes differentiation with respect to the scalar argument:

$$\psi_n'(t) \triangleq \frac{\mathrm{d}}{\mathrm{d}t}\,\psi_n(t). \tag{61}$$

Under on-policy sampling, evaluating the scalar coefficient $\psi_n'(\cdot)$ at the current iterate yields the gradient-equivalence established in the main text (see Theorem in Section 5.2 and Section C). A common choice $\psi_2(t)=\frac{1}{2}t^2$ recovers the squared log-ratio penalty. The total objective is then $\mathcal{L}_{\mathrm{reward,MC}}-\beta\cdot\mathcal{L}_{k_n\text{ as loss,MC}}$, with $\pi_{\mathrm{ref}}$ fixed (no gradients flow through it).

## E. Formal Analysis of the $k_3$ as loss Gradient Surrogate

The preceding section established that $k_1$ **in reward** and $k_2$ **as loss** are gradient-correct implementations of the RKL regularizer. A natural question is whether other formulations—particularly $k_3$ **as loss**, which is widely used in GRPO—also implement the same gradient. This section provides the formal analysis showing that they do not.

Specifically, we prove that $k_3$ **as loss** is a first-order, biased surrogate for the principled RKL gradient, and we rigorously characterize its three core deficiencies:

1. **Local bias**: The induced coefficient differs from the true RKL coefficient except at $\pi_\theta = \pi_{\text{ref}}$.

2. **Tail asymmetry**: The surrogate saturates when the policy over-covers the reference and diverges when it under-covers.

3. **Statistical instability**: The variance of the induced coefficient equals the chi-squared divergence, which can be unbounded.

**Setup and notation.** Throughout this section, we fix a prompt $x$ and consider on-policy samples $y \sim \pi_\theta(\cdot|x)$. To simplify notation, we define the probability ratio:

$$\delta(y) := \frac{\pi_{\text{ref}}(y|x)}{\pi_\theta(y|x)}. \tag{62}$$

The ratio $\delta$ captures how much the reference distribution favors action $y$ relative to the current policy: $\delta > 1$ means the reference assigns more probability to $y$ than the current policy (under-coverage), while $\delta < 1$ means the opposite (over-coverage). Our analysis compares the coefficient induced by $k_3$ **as loss** against the principled RKL gradient coefficient, $c^\star(y) = -\log \delta(y)$. All scalar coefficients multiplying the score function $\nabla_{\boldsymbol{\theta}} \log \pi_\theta(y|x)$ are treated as detached.

**Gradient-Equivalent Coefficient of $k_3$ as loss.** The $k_3$ **as loss** objective is given by:

$$\mathcal{J}_{k_3 \text{ as loss}}(\boldsymbol{\theta}) = \mathbb{E}_{x\sim\mathcal{D},\, y\sim\pi_\theta(\cdot|x)}\left[ \frac{\pi_{\text{ref}}(y|x)}{\pi_{\boldsymbol{\theta}}(y|x)} - 1 - \log \frac{\pi_{\text{ref}}(y|x)}{\pi_{\boldsymbol{\theta}}(y|x)} \right]. \tag{63}$$

Differentiating and evaluating the resulting scalar multiplier at the detached snapshot yields:

$$\nabla_{\boldsymbol{\theta}} \mathcal{J}_{k_3 \text{ as loss}}(\boldsymbol{\theta}) = \mathbb{E}_{x\sim\mathcal{D},\, y\sim\pi_\theta(\cdot|x)}\left[ \left(1 - \frac{\pi_{\text{ref}}(y|x)}{\pi_\theta(y|x)}\right) \nabla_{\boldsymbol{\theta}} \log \pi_\theta(y|x) \right]. \tag{64}$$

This confirms that $k_3$ **as loss** is gradient-equivalent (under on-policy sampling) to an 'in-reward' update with the following scalar coefficient:

$$c_{3'}(y) := 1 - \delta(y). \tag{65}$$

$$\mathcal{J}_{k_{3'} \text{ in reward}}(\boldsymbol{\theta}) = \mathbb{E}_{x\sim\mathcal{D},\, y\sim\pi_\theta(\cdot|x)}\left[ \left(1 - \frac{\pi_{\text{ref}}(y|x)}{\pi_\theta(y|x)}\right) \log \pi_{\boldsymbol{\theta}}(y|x) \right]. \tag{66}$$

The remainder of this section formally analyzes the deficiencies of this proxy, $c_{3'}$, when compared to the principled target, $c^\star = -\log \delta$.

**Lemma E.1** (First-order agreement and second-order bias)**.** *The proxy $1 - \delta$ is the first-order Taylor approximation of the principled coefficient $-\log \delta$ around $\delta = 1$. The approximation error (bias) is of second order:*

$$\text{Bias}(\delta) = (-\log \delta) - (1 - \delta) = \frac{1}{2}(\delta - 1)^2 - \frac{1}{3}(\delta - 1)^3 + O\big((\delta - 1)^4\big). \tag{67}$$

*Proof sketch.* The result is obtained by expanding $-\log \delta$ in a Taylor series at $\delta = 1$ and subtracting the term $(1 - \delta)$. $\qquad\square$

**Lemma E.2** (One-sided domination and asymmetric tails)**.** *For all $\delta > 0$, the proxy is a strict lower bound, $1 - \delta \leq -\log \delta$, with equality holding only at $\delta = 1$. Their tail behaviors are pathologically asymmetric:*

- ***Over-coverage*** *($\delta \to 0^+$): The proxy provides a weak, saturating restoring force ($\lim_{\delta\to0^+}(1 - \delta) = 1$), whereas the principled coefficient provides an unbounded penalty ($\lim_{\delta\to0^+}(-\log \delta) = +\infty$).*

- **Under-coverage** ($\delta \to \infty$): *The proxy induces an unbounded* linear *penalty ($\lim_{\delta \to \infty}(1 - \delta) = -\infty$), while the principled coefficient grows only logarithmically.*

*Proof sketch.* The inequality follows from the fundamental property $\log \delta \leq \delta - 1$. The limits are elementary. $\qquad \square$

**Theorem E.1** (Variance equals chi-squared divergence). *Assuming* $\mathrm{supp}(\pi_{ref}(\cdot|x)) \subseteq \mathrm{supp}(\pi_\theta(\cdot|x))$, *the proxy coefficient* $c_{3'}$ *has zero mean under the on-policy sampling distribution, and its variance is exactly the chi-squared divergence:*

$$\begin{aligned} \mathbb{E}_{y \sim \pi_\theta}[1 - \delta(y)] &= 0, \\ \mathrm{Var}_{y \sim \pi_\theta}[1 - \delta(y)] &= \chi^2(\pi_{ref}(\cdot|x) \parallel \pi_\theta(\cdot|x)). \end{aligned} \tag{68}$$

*If the support condition is violated, the variance is infinite.*

*Proof sketch.* $\mathbb{E}[\delta] = \sum_y \pi_\theta(y|x)\frac{\pi_{\mathrm{ref}}(y|x)}{\pi_\theta(y|x)} = 1$, thus $\mathbb{E}[1 - \delta] = 0$. The variance identity then follows directly from the definition of $\chi^2(p \parallel q)$. $\qquad \square$

**Corollary E.1.1** (Implication for stochastic gradient variance). *The variance of the stochastic gradient term induced by $k_3$* **as loss** *is directly governed by the chi-squared divergence, an often unstable metric:*

$$\mathbb{E}\left[\|(1 - \delta(y))\nabla_{\boldsymbol{\theta}} \log \pi_{\boldsymbol{\theta}}(y|x)\|^2\right] = \mathbb{E}\left[(1 - \delta(y))^2 \|\nabla_{\boldsymbol{\theta}} \log \pi_{\boldsymbol{\theta}}(y|x)\|^2\right]. \tag{69}$$

**Summary.** These results provide a rigorous, gradient-centric justification for the claims in the main text:

- The $k_3$ **as loss** formulation does *not* implement the true RKL gradient.

- It deploys a first-order proxy ($c_{3'} = 1 - \delta$) that is accurate only when the policy is very close to the reference ($\delta \approx 1$).

- Its pathological tail behavior—saturating at $+1$ in one direction while diverging to $-\infty$ in the other—introduces optimization challenges not present in the principled $k_1$ **in reward** or $k_2$ **as loss** formulations.

- Its variance equals the chi-squared divergence, which can be unbounded under distribution shift.

This analysis underscores the importance of selecting regularization losses based on their gradient properties, not merely their characteristics as value estimators.

# F. Derivation of an Alternative Regularizer: The MiniMax-01 Loss

The preceding analysis revealed that KL-based coefficients can be unbounded: the principled RKL coefficient $\log(\pi_\theta/\pi_{\mathrm{ref}})$ grows without bound as the policy diverges from the reference. While this provides strong regularization, it can also induce large gradient updates that destabilize training in some settings.

This section derives the MiniMax-01 loss (Li et al., 2025) as an alternative regularizer with a *bounded* coefficient. We show that it originates from a mean squared error (MSE) objective in probability space and fits naturally within our gradient-centric framework. The key advantage is that the induced coefficient $\pi_\theta - \pi_{\mathrm{ref}}$ is always bounded in $[-1, 1]$, providing a more conservative regularization force.

**Objective.** We minimize the full-vocabulary MSE between the policy and the reference:

$$\mathcal{J}_{\mathrm{MSE}}(\boldsymbol{\theta}) = \mathbb{E}_{x \sim \mathcal{D}}\left[\frac{1}{2}\sum_y \left(\pi_{\boldsymbol{\theta}}(y \mid x) - \pi_{\mathrm{ref}}(y \mid x)\right)^2\right]. \tag{70}$$

**On-policy gradient.** Differentiating Equation (70) with respect to $\boldsymbol{\theta}$, evaluating the scalar multiplier at the detached snapshot $\pi_\theta$ (our standard on-policy convention), and converting to the score-function form yields:

$$
\begin{aligned}
\nabla_{\boldsymbol{\theta}} \mathcal{J}_{\mathrm{MSE}}(\boldsymbol{\theta}) &= \mathbb{E}_{x \sim \mathcal{D}} \sum_y \left[ \frac{1}{2} \nabla_{\boldsymbol{\theta}} (\pi_{\boldsymbol{\theta}}(y|x) - \pi_{\mathrm{ref}}(y|x))^2 \right] \\
&= \mathbb{E}_{x \sim \mathcal{D}} \sum_y \underbrace{\left( \pi_\theta(y \mid x) - \pi_{\mathrm{ref}}(y \mid x) \right)}_{\text{scalar coefficient}} \nabla_{\boldsymbol{\theta}} \pi_{\boldsymbol{\theta}}(y \mid x) \\
&= \mathbb{E}_{x \sim \mathcal{D}} \sum_y \pi_\theta(y \mid x) \left( \pi_\theta(y \mid x) - \pi_{\mathrm{ref}}(y \mid x) \right) \nabla_{\boldsymbol{\theta}} \log \pi_{\boldsymbol{\theta}}(y \mid x) \\
&= \mathbb{E}_{x \sim \mathcal{D}, \, y \sim \pi_\theta(\cdot \mid x)} \left[ \left( \pi_\theta(y \mid x) - \pi_{\mathrm{ref}}(y \mid x) \right) \nabla_{\boldsymbol{\theta}} \log \pi_{\boldsymbol{\theta}}(y \mid x) \right].
\end{aligned}
\tag{71}
$$

The last line reveals that MSE regularization induces a score-function update whose scalar coefficient is the probability difference $\pi_\theta - \pi_{\mathrm{ref}}$.

**MiniMax-01 surrogate loss (Monte Carlo).** Using a on-policy sampler that draws $G$ responses $y^{(i,j)} \sim \pi_\theta(\cdot \mid x^{(i)})$ per prompt, the unbiased minibatch surrogate whose negative gradient recovers Equation (71) is

$$
\mathcal{L}_{\mathrm{MSE,MC(MiniMax\text{-}01)}}(\boldsymbol{\theta}) = -\frac{1}{NG} \sum_{i=1}^N \sum_{j=1}^G \left( \pi_\theta(y^{(i,j)} \mid x^{(i)}) - \pi_{\mathrm{ref}}(y^{(i,j)} \mid x^{(i)}) \right) \log \pi_{\boldsymbol{\theta}}(y^{(i,j)} \mid x^{(i)}).
\tag{72}
$$

This head shares the same in-reward score-function structure as our principled KL implementations: the coefficient does not carry gradients, and only $\log \pi_\theta$ contributes to the gradient.

**Key properties and implications.**

1. Bounded gradient coefficient. Since $0 \le \pi_\theta(y \mid x), \pi_{\mathrm{ref}}(y \mid x) \le 1$, the coefficient satisfies $-1 \le \pi_\theta(y \mid x) - \pi_{\mathrm{ref}}(y \mid x) \le 1$. This boundedness enhances stability against large or pathological updates, in contrast to the unbounded log-ratio used by KL (see Figure 1). This supports our recommendation in Section 5 to consider bounded alternatives when stability is paramount.

2. Symmetry in probability space. The MSE penalty is symmetric with respect to probability differences, providing more conservative corrections when policies diverge, compared to the logarithmic penalty of Reverse KL.

3. Off-policy compatibility. Owing to its in-reward form (where the coefficient does not carry gradients), this head is fully compatible with importance sampling and clipping, following the same correction rules as in Section G.

**Remark.** Consistent with our KL analysis, Equation (71) is obtained by evaluating scalar multipliers at the detached snapshot $\pi_\theta$ (on-policy). This expresses the regularizer in the same coefficient-times-score-function form, enabling direct comparison of the induced update dynamics.

# G. Off-Policy Correction for KL Regularization

All derivations so far have assumed *on-policy* sampling: the samples $y$ are drawn from the current policy $\pi_\theta$ at the moment of gradient computation. In practice, however, many RLHF implementations—particularly PPO with multiple gradient steps per rollout—operate in an *off-policy* setting where samples are drawn from a stale behavior policy $\pi_{\theta_k}$.

Off-policy updates require importance sampling (IS) and clipping for *both* the reward head and the KL head. When KL is implemented in the combined "in reward" form, the IS correction is inherited automatically from the PPO surrogate. However, when KL is implemented "as loss" in a decoupled form, the required IS/clipping on the KL term is easy to overlook, leading to biased gradients.

This section derives the principled correction and provides concrete guidance for both combined and decoupled objective structures.

### G.1. From On-Policy to Off-Policy Gradients

In the policy-gradient view, updates are driven by a scalar coefficient $c(x, y)$ (which does not carry gradients) multiplying the score function. The on-policy gradient estimator is:

$$\nabla_{\boldsymbol{\theta}} \mathcal{J}_c(\boldsymbol{\theta}) = \mathbb{E}_{x \sim \mathcal{D}, \, y \sim \pi_\theta(\cdot | x)} \big[ c(x, y) \, \nabla_{\boldsymbol{\theta}} \log \pi_{\boldsymbol{\theta}}(y \mid x) \big], \tag{73}$$

where $\pi_\theta$ is a detached snapshot numerically equal to $\pi_{\boldsymbol{\theta}}$ at the time of gradient evaluation. For samples drawn from a behavior policy $y \sim \pi_{\theta_k}(\cdot \mid x)$, an unbiased off-policy estimator requires IS, assuming the behavior policy has support over the sampled data ($\pi_{\theta_k}(y \mid x) > 0$):

$$\nabla_{\boldsymbol{\theta}} \mathcal{J}_c(\boldsymbol{\theta}) = \mathbb{E}_{x \sim \mathcal{D}, \, y \sim \pi_{\theta_k}(\cdot | x)} \Big[ \underbrace{\frac{\pi_\theta(y \mid x)}{\pi_{\theta_k}(y \mid x)}}_{\text{IS weight (no gradient)}} \; c(x, y) \, \nabla_{\boldsymbol{\theta}} \log \pi_{\boldsymbol{\theta}}(y \mid x) \Big]. \tag{74}$$

In practice, PPO replaces this detached IS weight with the gradient-carrying ratio $\rho_k(\boldsymbol{\theta}) = \frac{\pi_\theta(y|x)}{\pi_{\theta_k}(y|x)}$ (where gradients flow only through the numerator) and employs a clipped surrogate objective to reduce variance. For any scalar coefficient $c(x, y)$ (which does not carry gradients), the clipped objective to be maximized is:

$$\mathcal{J}_{c,\text{clipped}}(\boldsymbol{\theta}) = \mathbb{E}_{x \sim \mathcal{D}, \, y \sim \pi_{\theta_k}(\cdot | x)} \Big[ \min \Big( \rho_k(\boldsymbol{\theta}) \, c(x, y), \; \text{clip}\big( \rho_k(\boldsymbol{\theta}), 1 - \epsilon, 1 + \epsilon \big) c(x, y) \Big) \Big]. \tag{75}$$

### G.2. Correcting $k_n$ as loss by Converting to an In-Reward Head

A $\boldsymbol{k_n}$ **as loss** head is gradient-correct only on-policy. To adapt it for off-policy use, it must first be converted to its gradient-equivalent in-reward form. This is achieved by defining a detached (stop-gradient) coefficient $k_{n'}(x, y)$ that reproduces the on-policy gradient of the original loss. For a differentiable penalty $k_n\big(\pi_\theta(y|x), \pi_{\text{ref}}(y|x)\big)$, this coefficient is its derivative with respect to the policy's log-probability, evaluated at the *current* detached snapshot:

$$k_{n'}\big(\pi_\theta(y|x), \pi_{\text{ref}}(y|x)\big) := \frac{\partial}{\partial \log \pi_{\boldsymbol{\theta}}} \, k_n\big(\pi_{\boldsymbol{\theta}}(y|x), \pi_{\text{ref}}(y|x)\big)\Big|_{\pi_{\boldsymbol{\theta}} = \pi_\theta}. \tag{76}$$

This conversion precisely aligns with the theoretical equivalences established in the main text:

- **Principled $k_2$ as loss**: $k_2 = \frac{1}{2}(\log \pi_\theta - \log \pi_{\text{ref}})^2 \; \Rightarrow \; k_{2'} = \log \pi_\theta - \log \pi_{\text{ref}}$ (the $k_1$ in reward coefficient).

- **Proxy $k_3$ as loss**: $k_3 = \frac{\pi_{\text{ref}}}{\pi_\theta} - 1 - \log \frac{\pi_{\text{ref}}}{\pi_\theta} \; \Rightarrow \; k_{3'} = 1 - \frac{\pi_{\text{ref}}}{\pi_\theta}$ (the $k_{3'}$ in reward coefficient).

Once expressed as a scalar coefficient $k_{n'}(x, y)$ (which does not carry gradients), the KL head is handled off-policy exactly like any other score-function head via Equation (75). In PPO with multiple epochs per batch, $\boldsymbol{k_{n'}}$ should be computed once using the rollout policy $\pi_{\theta_k}$ and held fixed across all epochs, consistent with standard PPO treatment of advantage estimates. The importance sampling ratio $\rho_k(\boldsymbol{\theta}) = \pi_\theta / \pi_{\theta_k}$ still updates with the policy across epochs, providing the necessary off-policy correction.

### G.3. Two Principled Off-Policy Integration Strategies

With the correctly derived coefficient $\boldsymbol{k_{n'}}$ in hand, there are two principled ways to integrate it into the PPO objective, mirroring the on-policy discussion in Section 4.

**1. Combined Form (Single Clipped Head).** Merge the reward advantage and the KL coefficient *before* applying the PPO machinery:

$$A_{\text{combined}}(x, y) := r(x, y) - \beta \, k_{n'}\big(\pi_\theta(y|x), \pi_{\text{ref}}(y|x)\big). \tag{77}$$

The clipped surrogate is then applied to this combined head:

$$\mathcal{J}_{\text{RLHF}}(\boldsymbol{\theta}) = \mathbb{E}_{y \sim \pi_{\theta_k}} \Big[ \min \Big( \rho_k(\boldsymbol{\theta}) \, A_{\text{combined}}, \; \text{clip}(\rho_k(\boldsymbol{\theta}), 1 - \epsilon, 1 + \epsilon) \, A_{\text{combined}} \Big) \Big]. \tag{78}$$

This is a robust and straightforward approach, as IS and clipping are consistently applied to both components. For correct PPO semantics, form $A_{\text{combined}}$ prior to any baseline subtraction or normalization. This preserves the trade-off set by $\beta$, which would be distorted by shifting or rescaling the KL component.

**2. Decoupled Form (Two Clipped Heads).** Maintain separate reward and KL objectives, each with its own IS correction and clipping scheme:

$$\mathcal{J}_{\text{reward}}(\boldsymbol{\theta}) = \mathbb{E}_{y \sim \pi_{\theta_k}} \left[ \min \left( \rho_k(\boldsymbol{\theta}) \, r(x, y), \; \text{clip}(\rho_k(\boldsymbol{\theta}), \, 1 - \epsilon_1, \, 1 + \epsilon_2) \, r(x, y) \right) \right], \tag{79}$$

$$\mathcal{J}_{\text{KL}}(\boldsymbol{\theta}) = \mathbb{E}_{y \sim \pi_{\theta_k}} \left[ \min \begin{pmatrix} \rho_k(\boldsymbol{\theta}) \, k_{n'}\big(\pi_\theta(y|x), \pi_{\text{ref}}(y|x)\big), \\ \text{clip}\big(\rho_k(\boldsymbol{\theta}), \, 1 - \epsilon, \, 1 + \epsilon\big) \, k_{n'}\big(\pi_\theta(y|x), \pi_{\text{ref}}(y|x)\big) \end{pmatrix} \right]. \tag{80}$$

The final objective to maximize is:

$$\mathcal{J}_{\text{RLHF}}(\boldsymbol{\theta}) \; := \; \mathcal{J}_{\text{reward}}(\boldsymbol{\theta}) \; - \; \beta \, \mathcal{J}_{\text{KL}}(\boldsymbol{\theta}). \tag{81}$$

This decoupled design affords greater flexibility, such as using asymmetric clipping for the reward head (e.g., $\epsilon_2 > \epsilon_1$) to accelerate learning, while retaining conservative, symmetric clipping for the KL head to ensure stable regularization. Baselines or normalization should be applied only to $A_{\text{reward}}$, not to $k_{n'}(x, y)$, to avoid implicitly altering the regularization strength $\beta$.

**Implementation Notes.**

- **Token vs. Sequence Level:** Our derivations use sequence-level probabilities. In token-level PPO, it is often more stable to compute the ratio as $\rho_k = \exp\left( \sum_t \log \pi_{\boldsymbol{\theta}}(y_t \mid \cdot) - \sum_t \log \pi_{\theta_k}(y_t \mid \cdot) \right)$ and apply clipping at the sequence level; per-token clipping can be overly conservative.

- **Masking Consistency:** Sum log-probabilities only over action tokens that contribute to the reward/KL (exclude prompt, padding, or masked tokens) to keep $\rho_k$ aligned with the heads being optimized.

- **Numerical Stability and Support:** Ensure $\pi_{\theta_k}(y \mid x) > 0$ for all sampled $(x, y)$ and consider numerically capping $\rho_k$ to prevent overflows under extreme ratios.

- **Adaptive $\beta$:** If targeting a desired KL via an adaptive schedule, update $\beta$ outside the gradient path (detached) and avoid mixing it with advantage normalization; adaptation is orthogonal to IS/clipping and works for both combined and decoupled forms.

**Sign Convention.** We present objectives for *maximization*. Implementations that *minimize* a loss should negate these expressions, e.g., by minimizing $-\mathcal{J}_{\text{combined}}$ or $-(\mathcal{J}_{\text{reward}} - \beta \, \mathcal{J}_{\text{KL}})$.

# H. Visualization of KL Regularization Gradient Coefficients

The theoretical analysis in Section 5 showed that different KL implementations induce different scalar coefficients multiplying the score function. Visualizing these coefficients as a function of the policy's log-probability provides valuable intuition about their behavior.

This section provides the Python code used to generate Figure 1. The plot compares four coefficient functions:

- The principled RKL coefficient $\log(\pi_\theta/\pi_{\text{ref}})$ (achieved by $k_1$ **in reward** and $k_2$ **as loss**);

- The biased GRPO coefficient $1 - \pi_{\text{ref}}/\pi_\theta$ (induced by $k_3$ **as loss**);

- The bounded MiniMax-01 coefficient $\pi_\theta - \pi_{\text{ref}}$;

- The constant coefficient 1 (induced by $k_1$ **as loss**, which provides no regularization).

The visualization highlights the linear behavior of the principled RKL coefficient versus the asymmetric saturation/unbounded growth of the $k_3$ surrogate in the tails.

```python
import torch
import matplotlib.pyplot as plt

# --- Plotting Style ---
```

```python
plt.style.use('seaborn-v0_8-whitegrid')
plt.rcParams.update({
    "text.usetex": False,  # Disable LaTeX rendering
    "font.family": "serif",  # Use a generic serif font
    "font.serif": ["Times New Roman"],  # Specify Times New Roman as the serif font
    "font.size": 14,
    "axes.labelsize": 16,
    "legend.fontsize": 12,
    "xtick.labelsize": 12,
    "ytick.labelsize": 12,
})

# --- Data Generation ---
log_pi_actor = torch.linspace(-5, 0, steps=400)
pi_actor = torch.exp(log_pi_actor)

pi_ref_val = 0.25
log_pi_ref = torch.log(torch.tensor(pi_ref_val))

# --- Coefficients Calculation ---
coeff_k1_loss = torch.ones_like(log_pi_actor)
coeff_k1_reward = log_pi_actor - log_pi_ref
coeff_k3_loss = 1 - pi_ref_val / pi_actor
coeff_minimax = pi_actor - pi_ref_val

# --- Plotting ---
plt.figure(figsize=(10, 6.5))

plt.plot(log_pi_actor, coeff_k1_reward,
         label=r'$\log\pi_{\theta} - \log\pi_{\text{ref}}$ ($k_1$ in reward / $k_2$ as
             loss) - Principled',
         color='#808000', linewidth=3, zorder=10)

plt.plot(log_pi_actor, coeff_k3_loss,
         label=r'$1 - \pi_{\text{ref}}/\pi_{\theta}$ ($k_{3^{\prime}}$ in reward / $k_3$
             as loss) - Biased Approximation',
         color='Firebrick', linestyle='--', linewidth=2)

plt.plot(log_pi_actor, coeff_minimax,
         label=r'$\pi_{\theta} - \pi_{\text{ref}}$ (MiniMax-01)',
         color='RoyalBlue', linestyle='-.', linewidth=2)

plt.plot(log_pi_actor, coeff_k1_loss,
         label=r'$1$ ($k_1$ as loss) - Zero Expected Gradient',
         color='Gray', linestyle=':', linewidth=2)

plt.axvline(x=log_pi_ref.item(), color='black', linestyle='--', linewidth=1,
            label=r'$\log\pi_{\theta} = \log\pi_{\text{ref}}$')

plt.xlabel(r'Actor Log-Probability: $\log \pi_{\theta}(y|x)$')
plt.ylabel(r'Coefficient of Score Function')
plt.title(r'Comparison of KL Regularization Coefficients ($\pi_{\text{ref}}=0.25$)',
    fontsize=18)
plt.legend(loc='upper left')

plt.ylim(-4, 4)
plt.xlim(-5, 0)

plt.tight_layout()
plt.savefig('comparison_kl_regularization_coefficients.png', dpi=300, bbox_inches='tight')
plt.show()
```

*Listing 1.* Python code to generate the comparison plot of KL gradient coefficients.

# I. On the Statistical Instability of the $k_3$ Value Estimator

The main text argues that KL implementations should be evaluated by their gradient properties, not their value-estimation properties. However, even when viewed purely as a value estimator, $k_3$ has significant limitations that are often overlooked.

This section analyzes the statistical properties of the $k_3$ value estimator. We show that its advertised "unbiasedness" relies on regularity conditions—specifically, absolute continuity of the reference with respect to the sampling distribution—that can fail in practice. When these conditions are violated, $k_3$ can be biased, and its variance can become infinite. These failure modes provide additional justification for preferring the principled $k_1$ **in reward** or $k_2$ **as loss** formulations.

## I.1. Precondition for Unbiasedness

An estimator is unbiased if its expectation equals the true value. For $k_3$:

$$\mathbb{E}_q[k_3] = \mathbb{E}_q[\delta(x) - 1 - \log \delta(x)] = (\mathbb{E}_q[\delta(x)] - 1) + D_{\mathrm{KL}}(q \parallel p) \tag{82}$$

For $k_3$ to be unbiased, it is necessary that $\mathbb{E}_q[\delta(x)] = 1$. This condition is met if $p$ is absolutely continuous with respect to $q$ ($p \ll q$), which means that the support of $p$ must be contained within the support of $q$.

The condition of a finite KL divergence ($D_{\mathrm{KL}}(q \parallel p) < \infty$) is **not sufficient** to guarantee unbiasedness. For example, let $q$ be the uniform distribution in $[0, 1]$ and $p$ be the uniform distribution on $[0, 2]$.

- The KL divergence $D_{\mathrm{KL}}(q \parallel p) = \int_0^1 1 \cdot \log(\frac{1}{0.5}) dx = \log 2$, which is finite.

- However, $\mathbb{E}_q[\delta(x)] = \int_0^1 1 \cdot \frac{p(x)}{q(x)} dx = \int_0^1 \frac{0.5}{1} dx = 0.5$.

- The estimator expectation is therefore $\mathbb{E}_q[k_3] = (0.5 - 1) + \log 2 = \log 2 - 0.5$, which is biased.

## I.2. Infinite Variance and the Chi-Squared Divergence

The variance of $k_3$ is dominated by the second moment of the importance ratio, $\mathbb{E}_q[\delta(x)^2]$. This term is directly related to the Chi-squared divergence.

When $p \ll q$, the identity holds: $\chi^2(p \parallel q) = \mathbb{E}_q[(\delta(x) - 1)^2] = \mathbb{E}_q[\delta(x)^2] - 1$. If $p$ is not absolutely continuous with respect to $q$ ($p \not\ll q$), $\chi^2(p \parallel q)$ is defined to be infinite.

Therefore, the variance of $k_3$ will be infinite if $\mathbb{E}_q[\delta(x)^2]$ is infinite. This occurs if $p \not\ll q$ or if $p \ll q$ but the tails of $q$ are sufficiently lighter than the tails of $p$. While the divergence of $\mathbb{E}_q[\delta(x)^2]$ is the primary cause of instability, the finiteness of $\mathrm{Var}(k_3)$ also technically requires the finiteness of $\mathbb{E}_q[(\log \delta(x))^2]$.

## I.3. The Gaussian Case and an Empirical Demonstration

For two Gaussian distributions, $p \sim \mathcal{N}(\mu_p, \sigma_p^2)$ and $q \sim \mathcal{N}(\mu_q, \sigma_q^2)$, the variance of $k_3$ is finite if and only if $\sigma_q^2 > \sigma_p^2/2$. This condition illustrates that the sampling distribution $q$ must be sufficiently "wide" relative to the reference distribution $p$. This condition generalizes for multivariate Gaussians with covariance matrices $\Sigma_p$ and $\Sigma_q$. **The expectation $\mathbb{E}_q[r(x)^2]$ is calculated via an integral involving the ratio of two Gaussian probability densities. For this integral to converge (and thus for the variance to be finite), it is required that the matrix $2\Sigma_q - \Sigma_p$ be positive definite.**

This failure mode is empirically illustrated below, where a narrow Gaussian $q(x)$ ($\sigma_q = 0.2$) is used to estimate the KL divergence to a standard Gaussian $p(x)$ ($\sigma_p = 1$). This configuration violates the condition, since $0.2^2 \not> 1^2/2$.

```
import torch
import torch.distributions as dist

# p: reference distribution, q: sampling distribution
p = dist.Normal(loc=0, scale=1)
q = dist.Normal(loc=0.1, scale=0.2) # A narrow distribution where Var[k3] is infinite

# Sample from the narrow distribution q
x = q.sample(sample_shape=(10_000,))
```

```
11 # Ground truth KL divergence D_KL(q || p)
12 true_kl = dist.kl_divergence(q, p)
13
14 # Compute the log-ratio log(p(x)/q(x))
15 log_r = p.log_prob(x) - q.log_prob(x)
16 r = torch.exp(log_r)
17
18 # Define estimators
19 k1 = -log_r
20 k2 = log_r.pow(2) / 2
21 k3 = r - 1 - log_r
22
23 # --- Code to generate output ---
24 print(f"True KL Divergence: {true_kl:.4f}\n")
25 print("Estimator         | Sample Mean    | Sample Std. Dev.")
26 print("------------------|---------------|------------------")
27 estimators = {"k1": k1, "k2": k2, "k3": k3}
28
29 for name, k in estimators.items():
30     mean = k.mean()
31     std = k.std()
32     print(f"{name:<17} | {mean:>13.4f} | {std:>16.4f}")
33
34 # --- Actual Output 1 ---
35 True KL Divergence: 1.1344
36
37 Estimator         | Sample Mean    | Sample Std. Dev.
38 ------------------|---------------|------------------
39 k1                |        1.1272 |           0.6912
40 k2                |        0.8742 |           0.6006
41 k3                |        0.8136 |           8.8244
42 # --- Actual Output 2 ---
43 True KL Divergence: 1.1344
44
45 Estimator         | Sample Mean    | Sample Std. Dev.
46 ------------------|---------------|------------------
47 k1                |        1.1336 |           0.6611
48 k2                |        0.8611 |           0.5210
49 k3                |        0.6817 |           4.1082
50 # --- Actual Output 3 ---
51 True KL Divergence: 1.1344
52
53 Estimator         | Sample Mean    | Sample Std. Dev.
54 ------------------|---------------|------------------
55 k1                |        1.1348 |           0.6709
56 k2                |        0.8689 |           0.4968
57 k3                |        0.6595 |           1.4925
58 # --- Actual Output 4 ---
59 True KL Divergence: 1.1344
60
61 Estimator         | Sample Mean    | Sample Std. Dev.
62 ------------------|---------------|------------------
63 k1                |        1.1256 |           0.6962
64 k2                |        0.8758 |           0.6263
65 k3                |        0.9772 |          26.7379
```

*Listing 2.* Code illustrating the high variance of the $k_3$ value estimator when the sampling distribution q(x) is too narrow.

The results illustrate the issue: across repeated trials, the sample standard deviation of $k_3$ can be substantially larger than that of $k_1$ and $k_2$, and the sample mean can deviate noticeably from the true KL value. This discrepancy is not systematic estimator bias, but rather sampling error caused by heavy-tailed variance, implying that far more samples may be needed for reliable estimation in such settings.

## J. Group Normalization Stability Issues

Beyond the gradient-level issues analyzed in the main text, GRPO's per-prompt group normalization can introduce a separate numerical stability problem. When the variance within a group of responses is very small, normalization can dramatically amplify tiny numerical differences, potentially destabilizing training.

This section describes the issue and proposes a simple fix: clipping the group standard deviation to prevent pathological amplification.

**The issue.** GRPO normalizes rewards within each group: for a prompt with $G$ responses and rewards $\mathbf{r} = \{r_1, \ldots, r_G\}$, the advantage is

$$A_i = \frac{r_i - \text{mean}_{\text{group}}(\mathbf{r})}{\text{std}_{\text{group}}(\mathbf{r})}. \tag{83}$$

**Stability issue.** When the within-group variance is very small (e.g., $\mathbf{r} = [0.99999, 1.00001, 0.99999, 1.00001]$), normalization can substantially amplify tiny numerical differences. For the above example, the resulting advantages become approximately $[-0.8660, 0.8660, -0.8660, 0.8660]$ (using the unbiased sample standard deviation), which can destabilize optimization by turning near-constant rewards into large-magnitude updates.

**Proposed solution.** Clip the standard deviation to prevent pathological amplification:

$$A_i = \frac{r_i - \text{mean}_{\text{group}}(\mathbf{r})}{\text{clip\_std}_{\text{group}}(\mathbf{r})},$$

$$\text{clip\_std}_{\text{group}}(\mathbf{r}) = \max\Big(\min(\text{std}_{\text{group}}(\mathbf{r}), \text{std}_{\max}), \text{std}_{\min}\Big). \tag{84}$$

Here, $\text{std}_{\min} > 0$ is a small floor that prevents exploding normalization when variance collapses, and $\text{std}_{\max}$ avoids under-normalization when variance is unusually large. In practice, setting $\text{std}_{\min}$ as a small constant relative to the reward scale (e.g., $10^{-1}$) may be effective.

**Why this matters beyond binary rewards.** Although binary 0/1 rewards in RLVR can sometimes mitigate extreme cases, more general regression reward models—such as those trained with Bradley–Terry (BT) losses—often produce continuous scores that may become highly concentrated (e.g., near 0 or 1) on easy or very hard prompts. In such regimes, within-group standard deviations can be arbitrarily small even when rewards are bounded in $[0, 1]$, and group normalization will over-scale negligible differences unless a variance floor (or clipping) is used. Therefore, std clipping is important not only for numerical stability but also to avoid over-amplifying noise when reward predictions saturate.

**Remark.** For reward scores bounded in $[0, 1]$, $\text{std}(\mathbf{r}) < 1$ always holds, but it can be orders of magnitude smaller than 1 in practice; the smaller the variance, the stronger the amplification effect from group normalization. Clipping $\text{std}_{\text{group}}(\mathbf{r})$ preserves the intended scale-invariance when variance is moderate, while guarding against instability when variance collapses.

## K. Forward KL vs. Reverse KL: A Gradient-Centric Reinterpretation

The main text focuses on the Reverse KL (RKL) divergence—the standard regularization objective in RLHF—and shows that $\boldsymbol{k_3}$ **as loss** is a biased first-order surrogate for the RKL gradient. An alternative perspective, recently noted by Tang & Munos (2025), is that $k_3$ as loss actually optimizes a Forward KL (FKL) objective, $D_{\text{KL}}(\pi_{\text{ref}} \| \pi_\theta)$.

This section refines this observation using our gradient framework and provides a deeper understanding of $\boldsymbol{k_3}$ **as loss**. We show that it is a *variance-reduced estimator* of the FKL gradient with an implicit baseline of $-1$. This explains two empirical observations:

- **Low variance near the reference**: When $\pi_\theta \approx \pi_{\text{ref}}$, the coefficient $1 - \delta$ fluctuates near zero, giving low variance.

- **Weak constraint under drift**: The FKL geometry is "mean-seeking" rather than "mode-seeking," imposing only a finite penalty for generating out-of-distribution tokens.

These properties make $k_3$ **as loss** behave well near the reference but provide weaker regularization when the policy drifts—consistent with the training instabilities observed in Section 6.

**Derivation of the FKL Gradient.** The Forward KL objective is defined as:

$$\mathcal{J}_{\text{FKL}}(\boldsymbol{\theta}) = \mathbb{E}_{x\sim\mathcal{D}}\Big[D_{\text{KL}}\big(\pi_{\text{ref}}(\cdot|x) \,\|\, \pi_{\boldsymbol{\theta}}(\cdot|x)\big)\Big] = \mathbb{E}_{x\sim\mathcal{D},\, y\sim\pi_{\text{ref}}(\cdot|x)}\big[\log \pi_{\text{ref}}(y|x) - \log \pi_{\boldsymbol{\theta}}(y|x)\big]. \tag{85}$$

Rewriting the gradient expectation using importance sampling over the current policy $\pi_\theta$ (to allow on-policy estimation) yields the standard FKL policy gradient:

$$\nabla_{\boldsymbol{\theta}}\mathcal{J}_{\text{FKL}}(\boldsymbol{\theta}) = \mathbb{E}_{x\sim\mathcal{D},\, y\sim\pi_\theta(\cdot|x)}\Big[ \underbrace{-\frac{\pi_{\text{ref}}(y|x)}{\pi_\theta(y|x)}}_{\text{Standard FKL coeff.}(-\delta)} \nabla_{\boldsymbol{\theta}}\log\pi_{\boldsymbol{\theta}}(y|x)\Big], \qquad \text{where } \delta := \frac{\pi_{\text{ref}}(y|x)}{\pi_\theta(y|x)}. \tag{86}$$

$k_3$ **as Loss: FKL with an Implicit Baseline.** Recalling Equation (18), the gradient induced by the $k_3$ as loss formulation uses the coefficient $1 - \delta$. Comparing this with the standard FKL coefficient in Equation (86) reveals an implicit decomposition:

$$\nabla_{\boldsymbol{\theta}}\mathcal{J}_{k_3 \text{ as loss}}(\boldsymbol{\theta}) = \mathbb{E}_{x\sim\mathcal{D},\, y\sim\pi_\theta(\cdot|x)}\Big[\big( \underbrace{(-\delta)}_{\text{Standard FKL}} - \underbrace{(-1)}_{\text{Implicit Baseline } b}\big)\nabla_{\boldsymbol{\theta}}\log\pi_{\boldsymbol{\theta}}(y|x)\Big]. \tag{87}$$

Although mathematically equivalent in expectation (since $\mathbb{E}[\nabla\log\pi]=0$), the implicit baseline $b=-1$ acts as a **control variate**:

- **Variance Reduction at Convergence:** Near the reference policy ($\pi_\theta \approx \pi_{\text{ref}}$), we have $\delta \approx 1$. The standard FKL estimator $-\delta$ fluctuates around $-1$ (high variance), whereas the $k_3$ coefficient $(1-\delta)$ fluctuates around 0. This explains why $k_3$ **exhibits low variance specifically** in the low-KL regime: it is effectively a zero-variance estimator of the FKL gradient at the optimum.

**Geometric Implications: Mean-Seeking vs. Mode-Seeking.** While the implicit baseline stabilizes estimation, optimizing FKL fundamentally alters the regularization geometry compared to the principled RKL:

- **RKL (Mode-seeking):** As derived in Section 5.2, RKL uses $-\log\delta$. As $\delta \to 0$ (policy places mass where reference does not), $-\log\delta \to \infty$. This creates a "barrier" that forces the policy to stay within the reference's support.

- **FKL (Mean-seeking):** The $k_3$ coefficient $1-\delta$ saturates at 1 as $\delta \to 0$. This "mean-seeking" behavior imposes only a finite penalty for generating out-of-distribution tokens. Consequently, under distribution shift, $k_3$ **as loss** fails to strongly penalize drift into regions unsupported by the reference model, consistent with the higher variability and weaker constraint compliance observed in our training curves.

In summary, $k_3$ **as loss** can be seen as a statistically coherent, baseline-corrected estimator of the FKL gradient, with favorable variance properties near the reference policy. However, its underlying geometry (mean-seeking and mode-covering) remains fundamentally different from that of RKL (mode-seeking). For RLHF applications where keeping the policy tightly constrained within the support of the reference model is a primary concern, the principled RKL implementations discussed in Section 5.2 may offer stronger and more reliable regularization.

$k_3$ **in reward: a mixed (and counterproductive) gradient.** A natural question is whether the nonnegative single-sample term $k_3$ can be used safely as an *in-reward coefficient* (i.e., a scalar multiplier without gradients). Our framework shows it cannot: unlike $k_1$, it does not induce the RKL gradient.

Let $\delta = \pi_{\text{ref}}(y|x)/\pi_\theta(y|x)$. By definition, $k_3 = \delta - 1 - \log\delta = k_1 + (\delta-1)$ with $k_1 = -\log\delta$. Therefore, under on-policy sampling,

$$\nabla_{\boldsymbol{\theta}}\mathcal{J}_{k_3 \text{ in reward}}(\boldsymbol{\theta}) = \mathbb{E}_{x\sim\mathcal{D},\, y\sim\pi_\theta(\cdot|x)}\Big[(k_1 + \delta - 1)\,\nabla_{\boldsymbol{\theta}}\log\pi_{\boldsymbol{\theta}}(y|x)\Big]$$

$$= \nabla_{\boldsymbol{\theta}}\mathcal{J}_{\text{RKL}}(\boldsymbol{\theta}) + \mathbb{E}_{x\sim\mathcal{D},\, y\sim\pi_\theta(\cdot|x)}\Big[\delta\,\nabla_{\boldsymbol{\theta}}\log\pi_{\boldsymbol{\theta}}(y|x)\Big], \tag{88}$$

where the constant term $-1$ drops by the zero-mean score identity. The remaining extra term rewrites as

$$\mathbb{E}_{x\sim\mathcal{D},\, y\sim\pi_\theta(\cdot|x)}\Big[\delta\,\nabla_{\boldsymbol{\theta}}\log\pi_{\boldsymbol{\theta}}(y|x)\Big] = \mathbb{E}_{x\sim\mathcal{D},\, y\sim\pi_{\text{ref}}(\cdot|x)}\Big[\nabla_{\boldsymbol{\theta}}\log\pi_{\boldsymbol{\theta}}(y|x)\Big] = -\nabla_{\boldsymbol{\theta}}\mathcal{J}_{\text{FKL}}(\boldsymbol{\theta}), \tag{89}$$

where the last equality follows from Equation (86). Hence,

$$\nabla_{\boldsymbol{\theta}}\mathcal{J}_{k_3\text{ in reward}}(\boldsymbol{\theta}) = \nabla_{\boldsymbol{\theta}}\mathcal{J}_{\text{RKL}}(\boldsymbol{\theta}) - \nabla_{\boldsymbol{\theta}}\mathcal{J}_{\text{FKL}}(\boldsymbol{\theta}). \tag{90}$$

Equivalently, using $k_3$ as a reward penalty corresponds to optimizing a *difference* of KL objectives, $\text{RKL} - \text{FKL}$, not their sum. In RLHF, this adds an *anti-FKL* component that can counteract the intended constraint and exacerbate instability. We therefore recommend using $k_1$ (or its gradient-equivalent surrogates) for reward shaping instead.

## L. Analysis of the DeepSeek-V3.2 Importance-Weighted KL Loss

The off-policy correction framework in Section G showed that $\boldsymbol{k_n}$ **as loss** implementations require explicit importance sampling (IS) and clipping when samples are drawn from a stale policy. A natural question is whether alternative IS constructions can avoid these complications.

DeepSeek-V3.2 (Liu et al., 2025a) proposes an IS-corrected $k_3$ estimator for off-policy KL regularization. Interestingly, PPO-style clipping is applied only to the reward head while the KL term is left **unclipped**. This section analyzes this construction through our gradient-centric lens and highlights a practical trade-off: under full differentiation, the estimator recovers the principled $k_1$ coefficient through cancellation; however, leaving the IS-weighted KL term unclipped can introduce numerical instability when the policy diverges from the behavior policy.

**The estimator.** DeepSeek defines the following IS-weighted $k_3$ loss (we use $\mathcal{L}_{\text{KL,DS}}$ to distinguish it from the true KL divergence $D_{\text{KL}}$):

$$\mathcal{L}_{\text{KL,DS}}(\boldsymbol{\theta}) = \frac{\pi_{\boldsymbol{\theta}}}{\pi_{\text{old}}}\left(\frac{\pi_{\text{ref}}}{\pi_{\boldsymbol{\theta}}} - \log\frac{\pi_{\text{ref}}}{\pi_{\boldsymbol{\theta}}} - 1\right). \tag{91}$$

Our gradient-centric analysis shows that, under full differentiation, this IS-corrected $k_3$ construction is mathematically equivalent to an in-reward $k_1$ coefficient, while leaving the IS-weighted KL term unclipped can be numerically fragile. We highlight two issues:

**Mathematical equivalence.** As derived below, minimizing the fully differentiated expectation of $\mathbb{E}[\rho(\boldsymbol{\theta})\cdot k_3(\boldsymbol{\theta})]$ analytically recovers the exact RKL gradient:

$$\nabla_{\boldsymbol{\theta}}\mathcal{J}_{\text{RKL}} = \mathbb{E}\left[\underbrace{\rho(\theta)\cdot\left(-\log\frac{\pi_{\text{ref}}}{\pi_\theta}\right)}_{\text{Evaluated Scalar Coefficient}}\cdot\nabla_{\boldsymbol{\theta}}\log\pi_{\boldsymbol{\theta}}\right]. \tag{92}$$

DeepSeek's estimator achieves this through cancellation of terms, effectively reconstructing the $k_1$ coefficient $-\log(\pi_\theta/\pi_{\text{ref}})$. Thus, under full differentiation, it is mathematically identical to the standard $\boldsymbol{k_1}$ **in reward** formulation (without clipping), while requiring additional computation.

**Practical trade-off with unclipped IS weights.** Leaving the IS-weighted KL term **unclipped** creates a practical trade-off:

- **Full-gradient case:** If the gradient path through $\rho(\boldsymbol{\theta})$ is preserved (so that Part A + Part B terms are both present), the resulting coefficient scales with the potentially large importance weight $\rho(\theta)$. Without clipping, this can yield very large gradients when the policy diverges from $\pi_{\text{old}}$.

- **Detached-weight case:** If $\rho$ is detached for stability, the estimator drops the sampling-distribution gradient term and reduces to the biased GRPO direction $(1 - \pi_{\text{ref}}/\pi_\theta)$, losing the equivalence to the RKL gradient.

### L.1. Mathematical Derivation and Dilemma Analysis

We analyze the gradient of the KL estimator in Equation (91). Let $\pi_{\boldsymbol{\theta}}$ denote the trainable policy carrying gradients, and $\pi_{\text{old}}$ denote the behavior policy (detached).

**Full Gradient Derivation (Equivalence to $k_1$).** The objective function corresponding to Equation (91) is:

$$\mathcal{L}_{\text{KL,DS}}(\boldsymbol{\theta}) = \mathbb{E}_{y \sim \pi_{\text{old}}} \left[ \underbrace{\frac{\pi_{\boldsymbol{\theta}}(y)}{\pi_{\text{old}}(y)}}_{\rho(\boldsymbol{\theta})} \cdot \underbrace{\left( \frac{\pi_{\text{ref}}(y)}{\pi_{\boldsymbol{\theta}}(y)} - \log \frac{\pi_{\text{ref}}(y)}{\pi_{\boldsymbol{\theta}}(y)} - 1 \right)}_{k_3(\boldsymbol{\theta})} \right]. \tag{93}$$

Applying the product rule $\nabla_{\boldsymbol{\theta}}(f \cdot g) = g \nabla_{\boldsymbol{\theta}} f + f \nabla_{\boldsymbol{\theta}} g$:

$$\nabla_{\boldsymbol{\theta}}\big(\rho(\boldsymbol{\theta}) \cdot k_3(\boldsymbol{\theta})\big) = \underbrace{(\nabla_{\boldsymbol{\theta}} \rho(\boldsymbol{\theta})) \cdot k_3(\theta)}_{\text{Part A}} + \underbrace{\rho(\theta) \cdot (\nabla_{\boldsymbol{\theta}} k_3(\boldsymbol{\theta}))}_{\text{Part B}}. \tag{94}$$

Note that in the product rule expansion, the term not being differentiated is treated as an evaluated value (hence black $\theta$).

**Part A (from $\nabla_{\boldsymbol{\theta}} \rho$):** Using the log-derivative trick $\nabla_{\boldsymbol{\theta}} \rho(\boldsymbol{\theta}) = \rho(\theta) \cdot \nabla_{\boldsymbol{\theta}} \log \pi_{\boldsymbol{\theta}}$:

$$\text{Part A} = \rho(\theta) \cdot \underbrace{\left( \frac{\pi_{\text{ref}}}{\pi_\theta} - \log \frac{\pi_{\text{ref}}}{\pi_\theta} - 1 \right)}_{k_3 \text{ scalar value}} \cdot \nabla_{\boldsymbol{\theta}} \log \pi_{\boldsymbol{\theta}}. \tag{95}$$

**Part B (from $\nabla_{\boldsymbol{\theta}} k_3$):** Differentiating $k_3$ with respect to $\boldsymbol{\theta}$ yields the biased GRPO coefficient:

$$\nabla_{\boldsymbol{\theta}} k_3(\boldsymbol{\theta}) = \left( 1 - \frac{\pi_{\text{ref}}}{\pi_\theta} \right) \nabla_{\boldsymbol{\theta}} \log \pi_{\boldsymbol{\theta}}, \tag{96}$$

and therefore:

$$\text{Part B} = \rho(\theta) \cdot \left( 1 - \frac{\pi_{\text{ref}}}{\pi_\theta} \right) \cdot \nabla_{\boldsymbol{\theta}} \log \pi_{\boldsymbol{\theta}}. \tag{97}$$

**Total Gradient (Cancellation):** Summing Part A and Part B, the bias terms in the coefficients cancel perfectly:

$$\begin{aligned}
\nabla_{\boldsymbol{\theta}} \mathcal{L}_{\text{KL,DS}} &= \rho(\theta) \cdot \left[ \left( \frac{\pi_{\text{ref}}}{\pi_\theta} - \log \frac{\pi_{\text{ref}}}{\pi_\theta} - 1 \right) + \left( 1 - \frac{\pi_{\text{ref}}}{\pi_\theta} \right) \right] \cdot \nabla_{\boldsymbol{\theta}} \log \pi_{\boldsymbol{\theta}} \\
&= \rho(\theta) \cdot \underbrace{\left( - \log \frac{\pi_{\text{ref}}}{\pi_\theta} \right)}_{\text{evaluated } k_1 \text{ coefficient}} \cdot \nabla_{\boldsymbol{\theta}} \log \pi_{\boldsymbol{\theta}}.
\end{aligned} \tag{98}$$

**Conclusion.** Under full differentiation, this construction recovers the same score-function coefficient as $k_1$ **in reward** (up to the IS weight). In this sense, it does not provide a different regularization direction; it realizes the $k_1$ coefficient through cancellation within a more complex expression.

**A practical trade-off with unclipped IS weights.** DeepSeek-V3.2 separates this term and leaves it **unclipped**. This leads to the following trade-off:

**Case 1 (full gradient path).** If $\rho(\boldsymbol{\theta})$ is kept active (i.e., both Part A and Part B are included), the gradient magnitude scales with $\rho(\theta)$. When the policy diverges from $\pi_{\text{old}}$ (e.g., $\pi_\theta \gg \pi_{\text{old}}$), $\rho(\theta)$ can become large; without clipping, this can produce very large gradients and numerical instability.

**Case 2 (detached weight).** If $\rho$ is detached for stability, then $\nabla_{\boldsymbol{\theta}} \rho \equiv 0$ and Part A vanishes. The estimator reduces to Part B:

$$\nabla_{\boldsymbol{\theta}} \mathcal{J}_{\text{detached}} = \rho(\theta) \cdot \left( 1 - \frac{\pi_{\text{ref}}}{\pi_\theta} \right) \cdot \nabla_{\boldsymbol{\theta}} \log \pi_{\boldsymbol{\theta}}. \tag{99}$$

This recovers the **biased GRPO coefficient** $1 - \pi_{\text{ref}}/\pi_\theta$, rather than the principled $k_1$ coefficient.

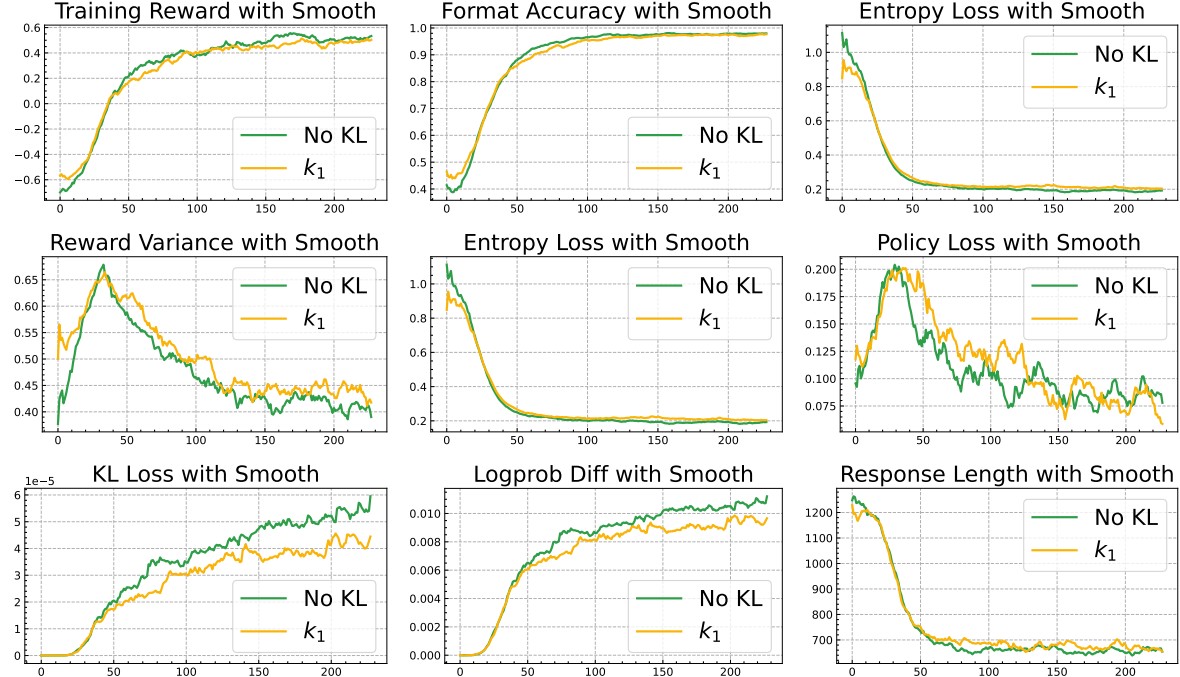

*Figure 5.* 1.5B-scale comparison of $k_1$ **as loss** versus no KL regularization. The gap is smaller than at 7B, but $k_1$ **as loss** still does not provide reliable regularization and can add stochasticity, consistent with its zero-mean, reference-independent expected gradient (Section 5).

**Summary.** This analysis highlights a practical tension when using unclipped, IS-weighted decoupled KL terms: preserving the full gradient path can lead to large updates when $\rho$ is large, while detaching $\rho$ changes the regularization direction. The principled approaches in Section 5.2—$k_1$ **in reward** with standard PPO correction, or the gradient-equivalent $k_2$ **as loss**—avoid this particular failure mode.

## M. Additional 1.5B-Scale Experiments

The main text presents experimental results at 7B scale. To assess the generality of our findings, this section provides complementary experiments at 1.5B scale using the Qwen2.5-Math-1.5B model.

The 1.5B experiments test the same hypotheses as the main text: (i) $k_1$ **as loss** should provide no meaningful regularization, and (ii) $k_2$ **as loss** should enforce a stronger constraint than $k_3$ **as loss**. Consistent with our theoretical analysis, we find that these patterns hold at 1.5B scale, though the differences between methods are somewhat less pronounced than at 7B.

Compared to the 7B-scale behavior in the main text (Figure 2), the 1.5B runs separate less. This matches our analysis: $k_1$ **as loss** mainly behaves as a zero-mean perturbation, so its qualitative effect is smaller at 1.5B and more pronounced at 7B in our runs.

**Weaker KL weight** ($\beta = 0.001$). We also conduct experiments at 1.5B scale with a weaker and more realistic KL weight ($\beta = 0.001$) to test the sensitivity of different surrogates under larger drift. As shown in Figure 7, the **KL Loss** panel reveals a clear loss spike for $k_3$ **as loss** (orange curve), while $k_2$ **as loss** (green curve) remains stable throughout training. This pattern mirrors the 7B results in Figure 4, confirming that the instability of the $k_3$ surrogate under distribution shift is consistent across model scales.

## N. Downstream Benchmark Performance

The main text (Table 3) reports a subset of downstream results for 7B models. This section provides the full evaluation across both model scales and additional benchmarks.

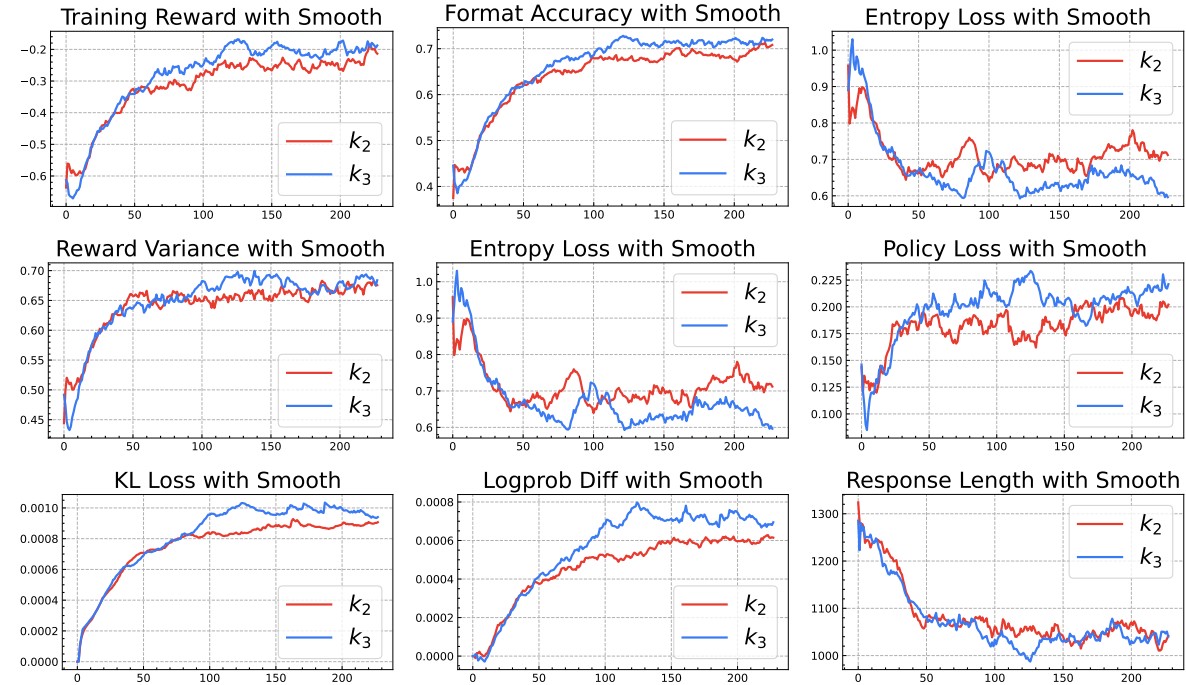

*Figure 6.* 1.5B-scale comparison of the principled $k_2$ **as loss** against its first-order surrogate $k_3$ **as loss**. Both variants constrain the policy, but $k_2$ **as loss** enforces a stronger, more stable regularization effect.

Table 4 summarizes downstream performance for 7B and 1.5B models across six math benchmarks (AIME 24/25, AMC, MATH-500, Minerva, Olympiad) and three general reasoning benchmarks (ARC-c, GPQA, MMLU-Pro). Several patterns emerge that are consistent with our theoretical analysis:

$k_1$ **as loss provides no regularization benefit.** Performance is close to the no-KL baseline, consistent with our analysis: its expected gradient vanishes and is independent of the reference model.

$k_2$ **as loss enforces a stronger constraint than $k_3$ as loss.** The principled $k_2$ **as loss** maintains tighter coupling to the reference, which in this setting correlates with lower downstream scores on both math benchmarks (AIME (Li et al., 2024), AMC, MATH-500 (Hendrycks et al., 2021)) and general reasoning benchmarks (ARC-c (Clark et al., 2018), GPQA* (Rein et al., 2024), MMLU-Pro (Wang et al., 2024)). In contrast, $k_3$ **as loss** allows larger divergence, which can yield higher reward but less stable training.

**Summary.** These results align with our theoretical analysis: $k_1$ **as loss** does not constrain; $k_2$ **as loss** implements the principled RKL and enforces a strong constraint; $k_3$ **as loss** is a weaker surrogate that permits larger drift.

# O. Statement on the Use of Large Language Models

We used LLMs solely for language polishing and editing. All retrieval of related work, algorithmic design, and theoretical derivations are carried out by the authors.

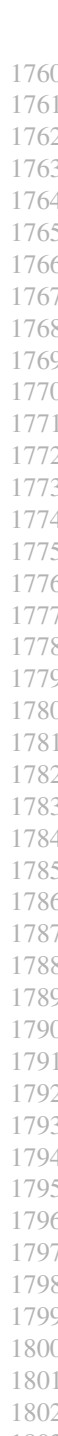

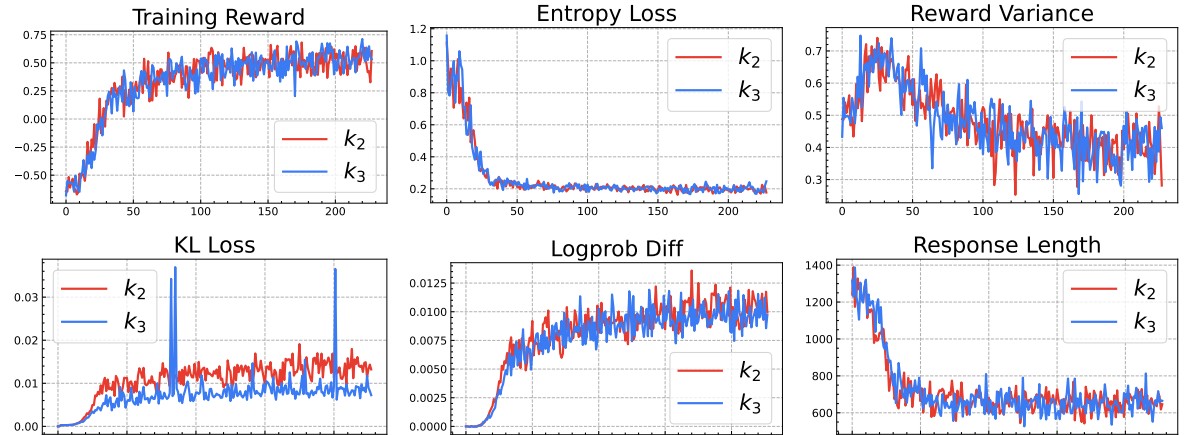

*Figure 7.* 1.5B-scale training diagnostics for $k_2$ **as loss** and $k_3$ **as loss** at a weaker KL weight ($\beta = 0.001$). The **KL Loss** panel (bottom-left) shows a visible spike for $k_3$ **as loss** that is absent for $k_2$ **as loss**, consistent with the 7B-scale results in Figure 4. This demonstrates that the $k_3$ surrogate's sensitivity to distribution shift persists across model scales.

*Table 4.* Main experiment results ($\beta = 0.5$) on math and general reasoning benchmarks based on **Qwen2.5-Math-7B** and **Qwen2.5-Math-1.5B**.

| Model | Math Reasoning Performance | | | | | | General Domain Reasoning Performance | | | |
|---|---|---|---|---|---|---|---|---|---|---|
| | AIME 24/25 | AMC | MATH-500 | Minerva | Olympiad | Avg. | ARC-c | GPQA* | MMLU-Pro | Avg. |
| Qwen2.5-Math-7B | 11.5/4.9 | 31.3 | 43.6 | 7.4 | 15.6 | 19.0 | 18.2 | 11.1 | 16.9 | 15.4 |
| RL w/o KL | 20.5/14.4 | 55.6 | 78.6 | 36.8 | 42.4 | 41.4 | 81.7 | 33.8 | 46.9 | 54.1 |
| RL w/. $k_1$ as loss | 19.1/11.6 | 56.0 | 80.6 | 40.8 | 43.0 | 41.8 | 79.7 | 29.8 | 45.1 | 51.5 |
| RL w/. $k_2$ as loss | 15.4/7.5 | 48.5 | 64.2 | 16.9 | 24.9 | 29.6 | 31.3 | 15.2 | 27.1 | 24.5 |
| RL w/. $k_3$ as loss | 19.0/7.3 | 48.9 | 65.4 | 18.8 | 29.0 | 31.4 | 29.6 | 19.2 | 27.7 | 25.5 |
| Qwen2.5-Math-1.5B | 7.2/3.6 | 26.4 | 28.0 | 9.6 | 21.2 | 16.0 | 3.5 | 4.0 | 2.5 | 3.3 |
| RL w/o KL | 12.5/4.8 | 43.7 | 66.8 | 28.3 | 31.9 | 31.3 | 43.7 | 19.2 | 23.1 | 28.7 |
| RL w/. $k_1$ as loss | 13.8/4.7 | 41.5 | 68.0 | 25.7 | 31.9 | 30.9 | 36.6 | 18.2 | 21.0 | 25.3 |
| RL w/. $k_2$ as loss | 7.0/5.5 | 35.2 | 52.8 | 14.7 | 29.0 | 24.0 | 7.8 | 7.6 | 4.9 | 6.8 |
| RL w/. $k_3$ as loss | 7.7/3.8 | 34.9 | 54.2 | 15.8 | 28.0 | 24.1 | 11.3 | 8.1 | 5.5 | 8.3 |

