# OpenReview forum: "Rethinking KL Regularization in RLHF: From Wrong Value Estimation to Correct Gradient Optimization"
_ICML.cc/2026/Conference — Submitted to ICML 2026_

### Official Review · Reviewer_3b9q · 2026-02-24

**Soundness:** 3
**Presentation:** 2
**Significance:** 2
**Originality:** 2
**Overall Recommendation:** 2
**Confidence:** 2

**Summary:**

The paper discusses gradient estimators for the KL-regularization term used in RLHF/RLVR objectives.

**Compliance With Llm Reviewing Policy:**

Affirmed.

**Final Justification:**

Please see ack.

**Key Questions For Authors:**

N/A

**Strengths And Weaknesses:**

Overall, the paper is closely aligned with existing discussions on the same issue (e.g., the recent note at https://arxiv.org/pdf/2506.09477). Even setting aside that recent reference, the discussion here seems to be fairly standard. Given this, I am not convinced that the paper meets the bar for an ICML publication. The material would likely be a better fit for a blog post.

---

> ### Author Rebuttal · Authors · 2026-03-30
>
> Thank you for the candid assessment. We agree that novelty is the central question. A closely related preprint by Tang & Munos [1] has partial thematic overlap with our theoretical discussion. We wish to note that our work — including publicly time-stamped manuscripts and open-source code — was disseminated well before the appearance of [1]. Due to double-blind constraints, we cannot disclose identifying details in this response; upon acceptance, we will provide a complete chronological record in the camera-ready version. Regardless of timeline, we will acknowledge the thematic overlap in the revision and ensure no wording overstates novelty on any single shared observation.
>
> That said, we believe the paper goes substantially beyond a blog-style restatement, for four reasons:
>
> 1. **A unified implementation framework.** The contribution is not just "a few gradient pitfalls." By introducing the equivalent-coefficient formula (Eq. 7), the paper maps every `in reward` and `as loss` implementation to a scalar coefficient on the score function, enabling direct comparison of different RLHF code paths under one objective lens. This framework also naturally distinguishes combined vs. decoupled integration patterns.
>
> 2. **First proof of exact RKL implementation and equivalence.** We provide the first rigorous gradient-level proof that the exact implementation of the standard `KL(π_θ ‖ π_ref)` is `k1 in reward`, and that it is gradient-equivalent to `k2 as loss` (Theorem 1). Based on this, we offer an objective-conditional practical recipe: if the target is RKL, use `k1 in reward` or `k2 as loss`; `k3 as loss` (used in GRPO) corresponds to a different regularizer — it is a first-order approximation with asymmetric tail behavior, not a faithful implementation of the same objective.
>
> 3. **Connection to current practice.** The paper also analyzes two issues commonly overlooked in practice: (a) in off-policy settings, `as loss` KL heads require explicit IS/clipping corrections, which are often omitted, leading to biased gradients; (b) DeepSeek-V3.2's IS-weighted correction to `k3` [2] appears to be a new KL implementation, but we prove in the Appendix that under full differentiation its gradient is exactly equivalent to `k1 in reward` — it effectively recovers the principled implementation through cancellation, rather than introducing a genuinely new method.
>
> 4. **A live, unresolved issue in the RLHF ecosystem.** This paper does not revisit a settled question. At the time of writing, `k3 as loss` remains the default KL implementation in major open-source RLHF frameworks (including OpenRLHF, verl, Slime, TRL/GRPO pipelines, and numerous community forks) and is reproduced in a wide range of production and research training runs. While DeepSeek — the original popularizer of the `k3 as loss` route via GRPO — has since applied an IS-based correction in DeepSeek-V3.2, the broader open-source ecosystem has not converged on corrected implementations. Critically, the issue extends well beyond math-reasoning benchmarks: any RLHF application that relies on KL regularization for training stability — including safety alignment, creative writing, and instruction following — is affected when the KL head silently implements a different, less stable regularizer than intended.
>
> Therefore, we agree that **merely pointing out** the single equivalence fact is not sufficient novelty. However, our contribution is broader and practically urgent: from a system-relevant RLHF implementation perspective, we systematically clarify which widely used sampled KL heads are gradient-correct under the standard objective, which are not, and why this matters — not only in theory, but for the large number of ongoing RLHF training runs built on these frameworks today. We sincerely hope you may reconsider the paper in that light.
>
> [1] Yunhao Tang and Rémi Munos, "On a Few Pitfalls in KL Divergence Gradient Estimation for RL," arXiv:2506.09477. https://arxiv.org/abs/2506.09477
>
> [2] Aixin Liu et al., "DeepSeek-V3.2: Pushing the Frontier of Open Large Language Models," arXiv:2512.02556. https://arxiv.org/abs/2512.02556

---

> > ### Author Rebuttal · Reviewer_3b9q · 2026-04-04
> >
> > Thank you for the response. I appreciate the clarification regarding the intended contribution of the paper. The rebuttal makes a stronger case that the work is not merely pointing out an isolated pitfall, but also provides a unified framework for comparing different KL implementations and connects this analysis to current RLHF practice.
> >
> > That said, I still have some reservation about the level of novelty. Even if some of the results are formalized more systematically here, the main conclusions appear fairly close to discussions that were already circulating in the community. As a result, while I agree the paper may be useful to practitioners, I am still uncertain whether the contribution is sufficiently novel for ICML. Therefore, I will maintain my current score, but lower my confidence. I am also willing to defer to the recommendation of the other reviewers.

---

> > > ### Author Response · Authors · 2026-04-04
> > >
> > > Thank you for the constructive re-evaluation and for recognizing the unified framework and its connection to RLHF practice.
> > >
> > > The remaining question concerns priority of contribution, and we would like to clarify: apart from the FKL interpretation of k3, which is attributed to Tang & Munos (already cited in our paper — and prior to that, we had contributed the characterization of k3 as a first-order biased approximation of RKL), all core results in this paper can be substantiated with timestamped manuscript and code records. We will provide the complete timeline upon acceptance; if any confidential channel is available before the final decision, we would also proactively submit this evidence.

---

### Official Review · Reviewer_M3pM · 2026-03-10

**Soundness:** 4
**Presentation:** 2
**Significance:** 3
**Originality:** 3
**Overall Recommendation:** 4
**Confidence:** 3

**Summary:**

This paper analyzes KL regularization in RLHF, showing that many implementations choose KL terms based on their value-estimation properties rather than their optimization behavior.
It proves that using $k_1$ directly as a loss produces zero expected gradient, while $k_1$ in reward (PPO-style) and $k_2$ as loss are gradient-equivalent and correctly implement Reverse KL regularization.
Experiments with 1.5B–7B models show that $k_2$ as loss provides better training stability than the $k_3$ (GRPO-style) approximation.

**Compliance With Llm Reviewing Policy:**

Affirmed.

**Key Questions For Authors:**

1. Please address the weaknesses.
2. The font size of Figure 1 is too small, making it difficult to read.
3. The evaluation focuses on Qwen2.5-Math models. Does this limit the generality of the conclusions? It is recommended to validate the findings on a wider range of models.
4. The $k_3$ as loss is a variance-reduced FKL estimator. FKL is mean-seeking, whereas RKL is mode-seeking. In writing or creative tasks where diversity is preferred over precision, does $k_3$’s mean-seeking bias actually provide a performance benefit over the $k_2$?

**Limitations:**

yes

**Strengths And Weaknesses:**

Strengths
- The paper provides a bridge between different KL implementations, clarifying why seemingly identical code changes lead to different optimization behaviors.
- Table 2 serves as a practical guide for researchers, clearly categorizing which methods are RKL-correct and providing specific recommendations.

Weaknesses
- The differences between methods are "somewhat less pronounced" at the 1.5B scale than at the 7B scale. More analysis on this point would be valuable.
- Unified font size and figure layout would improve readability.

---

> ### Author Rebuttal · Authors · 2026-03-30
>
> Thank you for the positive assessment and for focusing on the practical scope of the paper.
>
> **1.5B vs. 7B separation.** We view this as consistent with, rather than contradictory to, the theory. When policy drift is small (as in the 1.5B experiments), different KL heads induce more similar effective updates, so the gap is less pronounced. As model scale or optimization pressure increases, the coefficient mismatch becomes more prominent — which is why the 7B setting shows clearer separation. We will make this low-drift vs. larger-drift interpretation explicit, rather than implying the gap must always be large.
>
> **Presentation.** We agree that Figure 1 was too hard to read. We have redrawn Figure 1 with substantially improved readability; the updated figure is available at https://anonymous.4open.science/r/kl_rebuttal-1D0E/fig1_kl_coefficients.png .
>
> **Generality beyond Qwen2.5-Math.** Our theoretical analysis addresses sampled KL implementation under the standard reference-anchored RLHF objective and does not depend on a specific model architecture. To address the generality concern, we have conducted additional experiments on Qwen2.5-7B and 32B (both non-Math variants; full training curves and 32B full results are available in the anonymous artifact at https://anonymous.4open.science/r/kl_rebuttal-1D0E ). 7B downstream benchmark results:
>
> | Model | olympiad | minerva | aime | amc | aime25 | math avg | arc_c | gpqa | mmlu_pro | general avg |
> |---|---|---|---|---|---|---|---|---|---|---|
> | Qwen2.5-7B (base) | 23.0% | 10.3% | 7.8% | 27.6% | 4.1% | 19.5% | 15.1% | 2.0% | 21.1% | 12.7% |
> | β=0 | 40.7% | 36.4% | 14.1% | 45.3% | 8.8% | 36.9% | 89.2% | 32.8% | 55.0% | 59.0% |
> | k1 (β=0.5) | 40.4% | 35.7% | 12.2% | 44.5% | 6.4% | 35.7% | 89.5% | 36.4% | 54.3% | 60.1% |
> | k2 (β=0.5) | 30.1% | 13.6% | 9.1% | 34.2% | 4.8% | 24.7% | 36.8% | 4.5% | 26.7% | 22.7% |
> | k3 (β=0.5) | 27.1% | 12.9% | 8.2% | 32.3% | 5.3% | 23.8% | 37.2% | 5.1% | 26.7% | 23.0% |
> | k1 (β=0.001) | 41.9% | 34.6% | 13.6% | 45.6% | 6.7% | 36.3% | 89.5% | 36.9% | 54.6% | 60.3% |
> | k2 (β=0.001) | 40.0% | 36.4% | 11.8% | 44.7% | 5.4% | 35.9% | 88.9% | 36.9% | 54.1% | 60.0% |
> | k3 (β=0.001) | 41.5% | 36.4% | 12.7% | 43.5% | 4.3% | 35.4% | 89.8% | 31.8% | 54.1% | 58.5% |
>
> On 32B the patterns are consistent: at β = 0.5, k2 outperforms k3 (math avg 29.5% vs. 28.4%, general avg 40.4% vs. 38.5%); at β = 0.001, k2 also slightly outperforms k3 (math 44.6% vs. 43.8%, general 68.4% vs. 68.0%). Combined with the 7B results, this demonstrates that `k2 as loss` is the more robust choice across non-Math-specialized models and parameter scales, and that the findings are not limited to Qwen2.5-Math.
>
> **Could `k3 as loss` benefit creative/diversity-oriented tasks?** This is a reasonable question. More precisely, `k3 as loss` can be interpreted as an FKL gradient estimator with an implicit baseline (see Appendix D). Therefore, if a practitioner **intentionally** wants more mean-seeking / mode-covering regularization, then in some tasks that value coverage or diversity, this geometry may indeed be beneficial. However, this should be understood as **choosing a different regularizer**, not as better implementing the standard `KL(π_θ ‖ π_ref)`. Our claim is not that every non-RKL divergence is invalid; our claim is that if the target is the standard reference-anchored RKL regularization used in mainstream RLHF, then the faithful implementations remain `k1 in reward` or gradient-equivalent `k2 as loss`. One reason is precisely the mechanism we identify: RKL's mode-seeking geometry matches on-policy MC more naturally — it most strongly penalizes regions that the current policy already over-weights, and these are exactly the regions most likely to be sampled under `π_θ`, making it more direct and robust for controlling policy drift. By contrast, even when viewed through the FKL lens, `k3 as loss` still inherits the tail behavior and stability trade-offs associated with FKL updates. For safety-critical RLHF settings, where uncontrolled drift can produce harmful or out-of-distribution outputs, we therefore still view RKL as the better default; if practitioners truly want a different coverage behavior, that choice should be stated explicitly as an alternative objective — as done in recent work exploring non-RKL divergences [1] — rather than as a superior estimator of the same regularizer.
>
>
> [1] Chaoqi Wang, Yibo Jiang, Chenghao Yang, Han Liu, and Yuxin Chen, "Beyond Reverse KL: Generalizing Direct Preference Optimization with Diverse Divergence Constraints," ICLR 2024.

---

> > ### Author Rebuttal · Reviewer_M3pM · 2026-04-04
> >
> > Thanks for the authors' rebuttal and the additional experimental results provided. My primary concerns regarding the generalizability and the presentation of the paper have been addressed. I maintain my positive score.

---

> > > ### Author Response · Authors · 2026-04-04
> > >
> > > Thank you for the positive assessment and for confirming that your concerns have been addressed. Your constructive comments have been invaluable in refining our work. We will incorporate the supplementary 7B/32B results, the redrawn Figure 1, and the promised clarifications in the revision.

---

### Official Review · Reviewer_cxdG · 2026-03-12

**Soundness:** 4
**Presentation:** 4
**Significance:** 2
**Originality:** 2
**Overall Recommendation:** 4
**Confidence:** 2

**Summary:**

This paper analyzes the advantages and limitations of different KL-divergence estimators when used as the reward or loss from the perspective of coefficient-weighted policy gradients. Its main conclusions include that using $k_1$ as loss is unreasonable, that $k_1$ in reward is equivalent to $k_2$ as loss, and that $k_2$ as loss performs better than $k_3$ as loss. The authors support these findings with extensive experiments.

**Compliance With Llm Reviewing Policy:**

Affirmed.

**Final Justification:**

If this work indeed predates the existing discussions (such as the references mentioned in the authors’ response to reviewer 3b9q), then its positive role in advancing the community’s understanding of KL estimation deserves recognition. For this reason, I raise my score to 4.

**Key Questions For Authors:**

refine it: I noticed that in Table 3, $k_2$ as loss actually performs the worst. Does this suggest that, although $k_2$ as loss is recommended as the best KL-divergence estimator in the paper, it may not always have a more positive effect in practice? More broadly, since one of the paper’s central claims is that $k_2$ as loss yields the best estimator, how does this insight inform general RL training for LLMs?

**Strengths And Weaknesses:**

**Strengths:**
1. This paper is clear to read and understand.
2. The analysis for different KL-divergence estimators is solid supported by the rigorous mathematical derivation.
3. The investigation of KL-divergence estimators provides practically meaningful guidance for RL training of LLMs.

**Weaknesses:**
1. The analysis of KL divergence from the perspective of policy gradient is not especially novel, and the paper appears to primarily systematize existing understanding, such as why $k_1$ as loss fails.

---

> ### Author Rebuttal · Authors · 2026-03-30
>
> Thank you for the positive assessment and the substantive question.
>
> **On novelty.** We respectfully disagree with the characterization of this work as primarily systematizing existing understanding. The core mathematical results were not previously formalized in the literature: that `k1 as loss` has zero expected gradient (Section 5.1); that `k1 in reward` and `k2 as loss` are gradient-equivalent (Theorem 1); and that `k3 as loss` is an implicitly baselined FKL gradient estimator rather than a faithful RKL implementation (Section 5.3 + Appendix D). To our knowledge, no prior work analyzed `in reward` / `as loss`, `combined` / `decoupled`, and off-policy corrections together under one unified gradient-level framework. Moreover, the `k3 as loss` route introduced by GRPO continues to be widely adopted in major open-source frameworks such as Slime, verl, and TRL — these new results have direct and urgent practical relevance.
>
> **On `k2 as loss` scoring lowest in Table 3.** First, Table 3 (Qwen2.5-Math-7B, β=0.5) primarily contrasts {k2, k3} (effective regularization) vs. {k1 as loss, no-KL} (no effective constraint) — as stated in the paper, effective regularizers "trade some accuracy for stability." Between k2 and k3 specifically, the training reward curves are comparable (Section 6), and the downstream benchmark gap is relatively small (29.6 vs. 31.4). Notably, in the additional Qwen2.5-7B and 32B (non-Math) experiments we report in our response to Reviewer M3pM, k2 slightly outperforms or matches k3 in all settings, showing that the Table 3 pattern is specific to Qwen2.5-Math rather than a general weakness of k2. `k2 as loss` is indeed our recommended choice — because it **faithfully implements the standard RKL regularizer and is more stable in training** — while `k3 as loss` satisfies neither.
>
> The real distinction between the two lies in **stability**, not downstream scores: under weaker regularization (β = 0.001), `k3` exhibits a visible KL-loss spike (Figure 3) while `k2` does not. The root cause is that the `k3` coefficient has asymmetric tail behavior (Figure 1): it saturates at 1 on the π_θ > π_ref side (too weak), and diverges on the π_θ < π_ref side (too strong). This makes the effective constraint strength of `k3` dependent on the specific training trajectory, and the results are indeterminate — on Qwen2.5-Math, it manifests as weaker regularization and slightly higher scores; but in our additional Qwen2.5-7B and 32B (non-Math variant) experiments, k2 slightly outperforms or matches k3 at both β settings (e.g., 32B math 44.6% vs. 43.8%, general 68.4% vs. 68.0%). This confirms that the Table 3 pattern is a specific manifestation rather than a general weakness of k2 as loss. By contrast, the `k2` coefficient `log(π_θ/π_ref)` grows logarithmically on both sides, providing symmetric and predictable restoring force.
>
> Combining results across different models and scales: on Qwen2.5-Math-7B, k3 scores slightly higher than k2 (31.4 vs. 29.6, Table 3 in the paper), but on Qwen2.5-7B and 32B (non-Math), k2 is slightly better. This confirms that the effective constraint of k3 is indeterminate — it may manifest as weaker constraint and higher scores in some settings, but not in others. Meanwhile, `k2 as loss` consistently offers better stability across all settings and implements the correct objective. Our recommendation is therefore clear: `k2 as loss` is the more robust default.

---

> > ### Author Rebuttal · Reviewer_cxdG · 2026-04-03
> >
> > Thanks for the authors’ response. In fact, I was already familiar with several of the paper’s main conclusions from various blogs and prior literature, such as the ineffectiveness of using $k1$ ​ as loss, and the analysis of the $k2$ and $k3$ ​ estimators. However, after reading the authors’ response to reviewer 3b9q, I am not completely certain whether this work predates those blogs or discussions, especially since many related blogs and papers did not cite it. That said, based on the current information, I am inclined to believe that this work did come earlier, and I have therefore increased my score, in recognition of the authors’ positive contribution to advancing the community.

---

> > > ### Author Response · Authors · 2026-04-04
> > >
> > > Thank you for your positive feedback, for recognizing the technical value of this work, and for your inclination to acknowledge our priority. We would also like to note that the corresponding manuscript and code have in fact been cited by other works and have had impact in the community. All core results can be substantiated with timestamped manuscript and code records; to preserve anonymity, specific evidence will be provided upon acceptance.

---

### Official Review · Reviewer_HDMY · 2026-03-12

**Soundness:** 3
**Presentation:** 3
**Significance:** 2
**Originality:** 2
**Overall Recommendation:** 4
**Confidence:** 2

**Summary:**

This paper studies the relationship between using KL divergence in the reward function and incorporating KL regularization directly in the loss under different formulations. Based on this unified framework, the authors analyze several common ways KL terms are used in practice and provide recommendations for their application. These recommendations are validated through experiments on math evaluation benchmarks.

**Compliance With Llm Reviewing Policy:**

Affirmed.

**Final Justification:**

The work aims to provide a general framework for RL algorithms that incorporate KL divergence in either the loss or the reward, and it offers useful insights that could inform the design of future algorithms. The motivation and broader impact was not clear to me in the beginning because the choice of regularization terms are not randomly picked in classic RL field. However the author's rebuttal indicating this is the case for RLHF area as well as some popular codebases, so I think this work can contribute to the community.

**Key Questions For Authors:**

One aspect I would be particularly interested in is whether the unified framework could lead to a single practical recipe for using KL terms that works consistently across a range of modern RL algorithms or models. This seems like a potentially valuable outcome of the work that unifying different KL formulations and identifying a broadly applicable approach that performs well across evaluation benchmarks.

I am especially interested in the practical side for two reasons. First, the estimators themselves are largely known, and the methodological novelty mainly lies in the unifying perspective. Second, in many RL algorithms the way KL terms are propagated is closely tied to how the overall objective is formulated, which makes the choice of KL implementation often an empirical design decision. For this reason, it would be particularly compelling to see whether the proposed insights can translate into a clear empirical guideline that consistently improves performance across different algorithms and tasks.

**Limitations:**

Yes

**Strengths And Weaknesses:**

**Strength**:

The paper is well written and easy to follow. The authors do a good job improving readability by using different colors to distinguish trainable parameters from detached parameters. The work aims to provide a general framework for RL algorithms that incorporate KL divergence in either the loss or the reward, and it offers useful insights that could inform the design of future algorithms.

**Weakness**:

I am not entirely convinced by the starting point of analyzing the use of $k_n$ in the reward versus $k_n$ in the loss. In many RL algorithms, the KL term is introduced in a principled way that is closely tied to the objective formulation and theoretical guarantees, rather than by first adding a reward term and then choosing which KL gradient form to apply. For example, in TRPO [1] the KL divergence appears as a constraint, so gradients are typically not propagated through the KL term. In contrast, in SAC [2] the KL term is incorporated directly into the objective (e.g., through soft policy evaluation), where it is naturally differentiated during optimization.

From this perspective, it is not immediately clear how broadly the recommended recipes apply to existing RL implementations. For instance, I am not aware of commonly used RL algorithms that employ $k_1$ directly as a loss term. In addition, importance sampling is often used to obtain unbiased estimates, or surrogate objectives are introduced for improved empirical performance.

If the paper aims to provide practical guidance on how KL terms should be used, it could be beneficial to further emphasize empirical validation. In particular, presenting a clear and practical recipe for KL usage and evaluating it across a broader range of tasks could strengthen the paper. It would also be helpful to demonstrate consistent improvements over strong baselines (e.g., GRPO-style methods). At present, the empirical results do not appear to clearly show that the recommended KL implementation consistently improves evaluation performance.

**Additional Questions and Issues**:

L.82 (left col): It would be helpful to define $k_1,k_2,k_3$ earlier, as this would make the presentation easier to follow for readers.

L.118 (left col): The paper claims that $k_3$ may fail under distributional shift, with the counterexample involving different support between $p,q$. However, it seems that such support mismatch could affect other estimators as well. Could the authors clarify why this issue is specific to $k_3$, and whether similar concerns would also apply to $k_1,k_2$?

L.129 (right col): The paper mentions that the sample is not differentiable. Should this instead refer to the reward function being non-differentiable? If the reward function is known and the policy $\pi$ follows a Gaussian distribution, it seems that the reparameterization trick could be applied to obtain differentiable samples. Could the authors clarify this point?

Eq(7): I did not fully follow the purpose of this definition. It appears that $k_1$ used as a loss and $k_2$ used as a reward are gradient-equivalent, and similarly $k_3$ as a loss and $k_{3'}$ as a reward are also gradient-equivalent. Could the authors clarify what conclusion or insight is intended from establishing these equivalences?

Eq(8,9): Could the authors clarify the motivation for defining the combined and decoupled forms? These terms appear primarily in Table 1 and are not extensively explained elsewhere in the paper. It also seems that KL in reward would naturally correspond to the combined form, while KL in loss would correspond to the decoupled form due to how gradients propagate. If this interpretation is correct, it might be helpful to simplify or better motivate these terminologies to make the presentation easier to follow.

Table1: I find it difficult to see how the PPO loss can be directly written in the form of Eq. (5), as the advantage function does not appear explicitly in that formulation. Additionally, it is unclear why REINFORCE would involve a KL term in this context. Some additional clarification on how these methods are mapped to the formulations in Table 1 would be helpful.

L.237 (right col): The statement that $k_1$ has increased variance is not entirely clear. Does this refer to the variance of the KL estimator itself? If the gradients induced by these formulations are equivalent and optimization only depends on the gradients, it is not obvious why the variance difference would matter in practice. Could the authors clarify this point?

L.238 (right col): It's claimed that the implementation of $k_2$ is more straightforward than that of $k_1$, but the reason for this is not entirely clear to me. Could the authors elaborate on why $k_2$ is considered easier to implement in practice?

L.276 (right col): The paper mentions that $k_2$ is more principled, but the reasoning behind this claim is not entirely clear to me. Could the authors elaborate on why $k_2$ is considered more principled compared to the other formulations?

L.316 (right col): The reference appears to point to Figure 3, but it seems that Figure 2 is intended. Please verify and correct the figure reference.

Table 3: I am not fully convinced that $k_3$ as a loss consistently performs better than $k_2$ as a loss on the evaluation metric. The paper suggests that allowing drift can lead to better scores but less stable training; however, the training loss curves appear similarly smooth. If the KL constraint is indeed important, it seems plausible that there could exist a regime where $k_2$ as a loss provides both improved training stability and competitive (or better) evaluation performance compared to $k_3$. Could the authors elaborate on this point?

---

> ### Author Rebuttal · Authors · 2026-03-31
>
> Thank you for the careful and constructive review. We respond to the Weakness and Additional Questions separately below.
>
> **Response to Weakness:**
>
> 1. **Scope (re: TRPO/SAC).** We agree TRPO/SAC introduce KL in principled, objective-specific ways. Our setting is analogous: in RLHF with target `KL(π_θ ‖ π_ref)`, the KL is directly differentiated (like SAC) but MC-estimated from samples, reducing `in reward` vs. `as loss` to: "does the coded gradient match the true derivative?" Our derivation is equally rigorous — from the exact RKL derivative, we prove `k1 in reward` is its exact implementation, gradient-equivalent to `k2 as loss` (Theorem 1). We will clarify this scope earlier.
>
> 2. **Practical recipe and applicability.** You mention not being aware of commonly used RL algorithms that employ `k1` directly as a loss — this is precisely our starting point. In RLHF practice, `k1`/`k2`/`k3` were originally proposed as KL **value estimators** (for monitoring), but were repurposed as **optimization losses** — two uses with different mathematical requirements. GRPO chose `k3 as loss` for its low variance as a value estimator; but our paper proves its gradient does not match RKL when backpropagated. This error persists in Slime, verl, TRL today. This directly answers your Key Question about whether the framework yields a universal recipe: **yes — for standard RKL, default to `k2 as loss` (or `k1 in reward`) on-policy, add IS/clipping off-policy, and avoid `k3 as loss`.** Experiments on Qwen2.5-Math and supplementary 7B/32B non-Math models consistently validate this. `k1 as loss` is not a practical recommendation but a **counterexample**: despite being an unbiased KL value estimator, backpropagating yields zero expected gradient — no regularization. Any surrogate must be checked for gradient correctness via Eq. (7).
>
> **Response to Additional Questions and Issues:**
>
> 4. **L.82.** Agreed; we will define `k1`, `k2`, `k3` earlier.
>
> 5. **L.118 (support mismatch).** Reasonable question — support mismatch affects all estimators. But `k3` is more fragile because it contains `π_ref/π_θ` explicitly: this ratio explodes when `π_θ → 0`, tying variance to χ² divergence (Section 5.3). `k1`/`k2` use `log(π_θ/π_ref)`, growing only logarithmically — much milder. When used as a loss, the same ratio propagates into the gradient coefficient, causing one-sided saturation/divergence (Section 5.3). We will distinguish "value-estimation variance" from "gradient-level pathology" more clearly.
>
> 6. **L.129.** Agreed, too broad. "Non-differentiable sampling" refers to discrete autoregressive LLM RLHF; it does not cover reparameterizable continuous-control settings. We will restrict the scope.
>
> 7. **Eq(7).** Clarification: the equivalence is `k2 as loss` ↔ `k1 in reward` (not the reverse). Eq. (7) computes `k_n' = ∂k_n/∂log π_θ` to get the equivalent `in reward` coefficient, then checks if it equals `k1`. Results: `k2'=k1` (✓); `k1'=1` (✗ zero-mean); `k3'=1−π_ref/π_θ` (✗ first-order).
>
> 8. **Eq(8,9).** These are two independent dimensions that can be freely combined (as noted in Table 1's caption: "`in reward` can also be used in a decoupled form"). PPO typically uses `in reward` + combined; GRPO uses `as loss` + decoupled. Off-policy: combined inherits PPO's IS/clipping; decoupled needs explicit IS/clipping on the KL head (commonly omitted; Appendix G). Will simplify.
>
> 9. **Table 1.** Eq. (5) is the general `in reward` form where a coefficient multiplies `log π_θ`. As noted in the paper, `r(x,y)` here is "typically a shaped advantage signal." In PPO's combined form, the KL coefficient is absorbed into the advantage: `A_comb = r(x,y) - β·k1` (Eq. 8) — this combined signal plays the coefficient role in Eq. (5), so the advantage IS there. The REINFORCE row maps REINFORCE++ (with KL), not vanilla REINFORCE. Will clarify in Table 1.
>
> 10. **L.237.** Both KL esitmator and KL loss
>
> 11. **L.238 / L.276.** "Straightforward": `k2` is a standalone differentiable loss head; `k1 in reward` requires a detached coefficient. "Principled": `k2 as loss` is an exact decoupled RKL implementation — not inherently superior to gradient-equivalent `k1 in reward`. Will revise.
>
> 12. **L.316.** Confirmed; should reference Figure 2. Will fix.
>
> 13. **Table 3.** `k3`'s coefficient is asymmetric (saturates on one tail, diverges on the other), making its constraint trajectory-dependent. Under weak regularization, `k3` shows KL-loss spikes while `k2` stays stable. In supplementary Qwen2.5-7B/32B (non-Math) experiments, `k2` at β=0.001 is more stable and matches or slightly outperforms `k3` — confirming your hypothesized regime exists.

---

> > ### Author Rebuttal · Reviewer_HDMY · 2026-04-03
> >
> > The authors addressed my concerns on the motivation of the paper as well as broader guidance. Therefore I'm willing to increase my rate.

---

> > > ### Author Response · Authors · 2026-04-04
> > >
> > > Thank you for confirming that your concerns have been addressed. Your detailed review has been invaluable in improving the paper. We will incorporate all promised improvements in the revision, including earlier definitions of k1/k2/k3 and the presentation fixes you suggested.

---

### Decision · Program_Chairs · 2026-04-30

**Decision:**

Reject

**Comment:**

Reviewers agree that, while the paper has some strengths, e.g., providing insights that may be useful to practitioners, the contributions of this paper are not sufficient for acceptance due to the following main concerns:

1. The main conclusions and messages of the paper appear fairly close to discussions that were already circulating in the community.
2. Missing citations to existing materials, including Hugging Face blogs, which expressed similar views. Meanwhile, reviewers also point out that these materials are themselves later in the timeline, It is difficult to tell whether these conclusions emerged independently as part of a broader community consensus or were influenced, at least indirectly, by this paper.

Overall, the originality of this paper is insufficient (or hard to verify) for acceptance.